# Intergenerational epigenetic inheritance of cancer susceptibility in mammals

**Bluma J Lesch[1†‡*], Zuzana Tothova[2,3], Elizabeth A Morgan[4], Zhicong Liao[5,6], Roderick T Bronson[7], Benjamin L Ebert[2,3], David C Page[1,8,9*]**

[1]Whitehead Institute, Cambridge, United States; [2]Department of Medicine, Division of Hematology, Brigham and Women's Hospital, Harvard Medical School, Boston, United States; [3]Broad Institute of MIT and Harvard, Cambridge, United States; [4]Department of Pathology, Brigham and Women's Hospital, Harvard Medical School, Boston, United States; [5]Department of Genetics, Yale School of Medicine, New Haven, United States; [6]Yale Cancer Center, Yale School of Medicine, New Haven, United States; [7]Department of Pathology, Tufts University School of Medicine and Veterinary Medicine, North Grafton, United States; [8]Department of Biology, Massachusetts Institute of Technology, Cambridge, United States; [9]Howard Hughes Medical Institute, Whitehead Institute, Cambridge, United States

*For correspondence:
bluma.lesch@yale.edu (BJL);
dcpage@wi.mit.edu (DCP)

Present address: †Department of Genetics, Yale School of Medicine, New Haven, United States; ‡Yale Cancer Center, Yale School of Medicine, New Haven, United States

Competing interests: The authors declare that no competing interests exist.

**Abstract** Susceptibility to cancer is heritable, but much of this heritability remains unexplained. Some 'missing' heritability may be mediated by epigenetic changes in the parental germ line that do not involve transmission of genetic variants from parent to offspring. We report that deletion of the chromatin regulator *Kdm6a* (*Utx*) in the paternal germ line results in elevated tumor incidence in genetically wild type mice. This effect increases following passage through two successive generations of *Kdm6a* male germline deletion, but is lost following passage through a wild type germ line. The H3K27me3 mark is redistributed in sperm of *Kdm6a* mutants, and we define approximately 200 H3K27me3-marked regions that exhibit increased DNA methylation, both in sperm of *Kdm6a* mutants and in somatic tissue of progeny. Hypermethylated regions in enhancers may alter regulation of genes involved in cancer initiation or progression. Epigenetic changes in male gametes may therefore impact cancer susceptibility in adult offspring.
DOI: https://doi.org/10.7554/eLife.39380.001

## Introduction

Intergenerational inheritance of epigenetic state may significantly impact disease susceptibility in animals, including humans. In addition to genetic information, male and female gametes transmit epigenetic regulatory information, in the form of covalent DNA modification, histone modification, and small RNAs, to the zygote at fertilization. Accumulating evidence indicates that the epigenetic information inherited from both maternal and paternal gametes can modulate gene expression and phenotype in progeny throughout the metazoan lineage (*Arico et al., 2011*; *Carone et al., 2010*; *Ciabrelli et al., 2017*; *Greer et al., 2011*; *Morgan et al., 1999*; *Siklenka et al., 2015*). In mammals, these transcriptional effects can manifest phenotypically as defects in early development (*Chong et al., 2007*; *Siklenka et al., 2015*) or as altered metabolic or behavioral states during adulthood (*Carone et al., 2010*; *Dias and Ressler, 2014*; *Ng et al., 2010*).

Consistent with these findings, there is mounting evidence that mature mammalian sperm carry an information-rich epigenome. Although the final stages of testicular sperm development involve extensive nuclear rearrangement, including widespread replacement of histones with protamines and nucleus-wide chromatin compaction, mammalian sperm are more than motile packages of DNA.

**eLife digest** Many diseases, such as certain cancers, run in families. Often, this is because several related individuals inherit a version of a gene that is faulty and causes the condition. But in a number of families with high rates of cancer, scientists are unable to pinpoint such disease-causing gene versions.

Instead, it is possible that individuals inherit healthy genes that are not read and interpreted correctly by the cells. This could be because of epigenetic changes, modifications that do not alter the genetic code but can instead turn genes on or off temporarily by adding or removing certain marks on the genetic information.

For a long time, researchers thought that epigenetic changes could not be passed from one generation to the next, but recent studies have revealed this is actually possible. However, it had never been shown that this could be associated with having a higher risk of developing cancer.

Now, Lesch et al. show that epigenetic changes passed from male mice to their offspring make these animals more likely to develop tumors than typical mice. In the experiments, mouse sperm were genetically engineered to have a mutation in a gene called *Kdm6a* (also called *Utx* by cancer researchers), which controls the placement of epigenetic marks. Male mice carrying a defective *Kdm6a* gene were then mated to normal females. The resulting offspring developed more tumors than mice produced from normal sperm, even though they inherited a normal copy of the *Kdm6a* gene from their mother. Lesch et al. also show that the offspring have epigenetic marks similar to the ones found in the mutant sperm. This may change whether genes that stop or promote tumor formation are switched on or off.

Certain cancer treatments work by targeting epigenetic changes. The results by Lesch et al. therefore call for more research into whether cancer patients exposed to these drugs could transmit these modifications if they have children soon after the end of their treatment. Ultimately, knowing more about how epigenetic changes are involved in inherited diseases may start to provide answers to families affected by cancer.

DOI: https://doi.org/10.7554/eLife.39380.002

Mature mammalian sperm exhibit relatively high levels of DNA methylation (*Monk et al., 1987*), contain populations of small RNAs (*Sharma et al., 2016*; *Siklenka et al., 2015*), and retain 5–10% of their histones (*Hammoud et al., 2009*; *Jung et al., 2017*), which bear extensive post-translational modifications (*Erkek et al., 2013*; *Hammoud et al., 2009*; *Luense et al., 2016*). Specific histone modifications at some loci in the male germ line have been conserved during mammalian evolution, implying a biologically important function (*Lesch et al., 2016*). Recent evidence suggests that mature mouse sperm also retain elements of large-scale three-dimensional genomic domains found in somatic cells (*Jung et al., 2017*; *Ke et al., 2017*).

Altered epigenetic states play a significant role in cancer pathogenesis, making cancer a strong candidate for sensitivity to intergenerational epigenetic effects. Many cancers are highly heritable, meaning that the presence of a given tumor or set of tumors in one individual increases the risk of developing the same tumors among close relatives (*Goldgar et al., 1994*; *Lichtenstein et al., 2000*). However, despite extensive genetic studies of many human cancers, a large fraction of this heritability remains unexplained by specific genetic mutations or variants (*Lichtenstein et al., 2000*; *Mucci et al., 2016*). Meanwhile, investigations into tumor biology have revealed that cancer is in part a disease of epigenetic dysregulation. Many tumors are characterized by significantly perturbed gene regulatory states and exhibit abnormal genome-wide histone methylation and DNA methylation profiles (*Dawson and Kouzarides, 2012*). Cancer genetics studies over the last decade have revealed that, when susceptibility genes can be identified, many encode chromatin regulators, implying that epigenetic changes contribute either to tumor initiation or tumor progression (*Baylin and Jones, 2011*; *Dawson and Kouzarides, 2012*).

*Kdm6a* (*Utx*) has been identified as a candidate tumor suppressor in cancer genetics studies. The KDM6A protein has histone demethylase activity toward lysine 27 on histone H3 (H3K27), as well as demethylase-independent functions in establishment of enhancer regions (*Hong et al., 2007*; *Lan et al., 2007*; *Wang et al., 2017*). *KDM6A* mutations are found in a variety of human cancers,

including multiple myeloma, renal cell carcinoma, bladder carcinoma, acute myeloid leukemia (AML), acute lymphoid leukemia (ALL), prostate cancer, and medulloblastoma (*Jones et al., 2012*; *Ntziachristos et al., 2014*; *Van der Meulen et al., 2014*; *van Haaften et al., 2009*). *KDM6A* has been mechanistically implicated as a tumor suppressor in ALL (*Ntziachristos et al., 2014*; *Van der Meulen et al., 2015*), AML (*Gozdecka et al., 2018*), and lung cancer (*Wu et al., 2018*), and also plays important developmental roles, especially in the heart (*Lee et al., 2012*; *Welstead et al., 2012*) and blood (*Beyaz et al., 2017*; *Thieme et al., 2013*). Because *KDM6A* functions primarily as a chromatin regulator, these studies imply that the epigenetic sequelae of *KDM6A* loss contribute to tumor initiation or progression.

Here, we delete *Kdm6a* specifically in the male germ line of the mouse, and evaluate gene regulatory and phenotypic effects in genetically wild type offspring. We find that offspring of *Kdm6a* male germline knockouts exhibit an increased incidence of tumors, and that this effect is enhanced when *Kdm6a* is deleted in the germ line in two successive generations. Because these effects are provoked by a single genetic lesion in the parent, we were able to define specific epigenetic changes resulting from this manipulation. We find widespread perturbation of H3K27 methylation state in the *Kdm6a* mutant male germ line, as well as increased levels of DNA methylation at specific loci. Some of the changes in DNA methylation observed in the mutant germ line are retained in somatic tissue of wild type progeny, and may affect the transcriptional regulation of genes involved in cancer susceptibility.

## Results

### Generation of wild type offspring from Kdm6a conditional germline knockout males

We designed a breeding strategy to produce genetically wild type male offspring from a *Kdm6a* mutant male germ line. We generated a germline-specific *Kdm6a* conditional knockout (*Kdm6a* cKO) in male mice by crossing a *Cre* recombinase driven by the *Ddx4* (*Mvh*) promoter (*Gallardo et al., 2007*) to a conditional allele of *Kdm6a* (*Welstead et al., 2012*). The *Cre* transgene is expressed in the prenatal germ line, and excision of the conditional allele is complete by the time postnatal spermatogenesis begins (*Hu et al., 2013*). *Kdm6a* is encoded on the X chromosome, so recombination of a single allele is sufficient to generate a complete knockout. Because developing spermatogenic cells are linked by cytoplasmic bridges until just before sperm are released, and therefore share cytoplasmic factors, loss of *Kdm6a* expression from the X chromosome affects both X- and Y-bearing spermatogenic cells even after meiosis (*Braun et al., 1989*) . Mating *Kdm6a* cKO males to wild-type females produced genetically wild type male offspring ('*Kdm6a* F1') and heterozygous female offspring (*Figure 1A*). *Cre*-negative littermates of *Kdm6a* cKO males were mated to age-matched wild type females, and the male offspring of these crosses were used as controls ('control F1'). Critically, *Kdm6a* F1 males are genetically wild type, but generated from a paternal germ line lacking KDM6A activity.

*Kdm6a* cKO males were fertile, produced male and female offspring at Mendelian ratios, and exhibited normal spermatogenesis (*Figure 1—figure supplement 1*). We confirmed the high efficiency of *Cre* recombinase activity (>97%) by genotyping the heterozygous female offspring of these crosses (*Figure 1—figure supplement 2*). Male *Kdm6a* and control F1s were housed with littermates and followed until natural death or morbidity requiring euthanasia, at which time all animals underwent complete necropsy. Mice surviving past 24 months of age were considered healthy survivors.

### Reduced survival and increased tumor incidence in Kdm6a F1 compared to control F1 males

We found that lifespans of *Kdm6a* F1 males were shorter than those of control F1s (*Figure 1B*, *Figure 1—figure supplement 3*). This effect was independent of whether the animal carried the *Ddx4-Cre* transgene (*Figure 1—figure supplement 4*) and of mode of death (*Figure 1—figure supplement 5*). There was no significant difference in weight and a mild reduction in body length for *Kdm6a* F1s compared to control F1s at the time of death (*Figure 1—figure supplement 6*).

We evaluated cumulative necropsy data to define pathological correlates of the difference in survival between *Kdm6a* F1s and control F1s (*Figure 1—figure supplement 7*). We found that *Kdm6a*

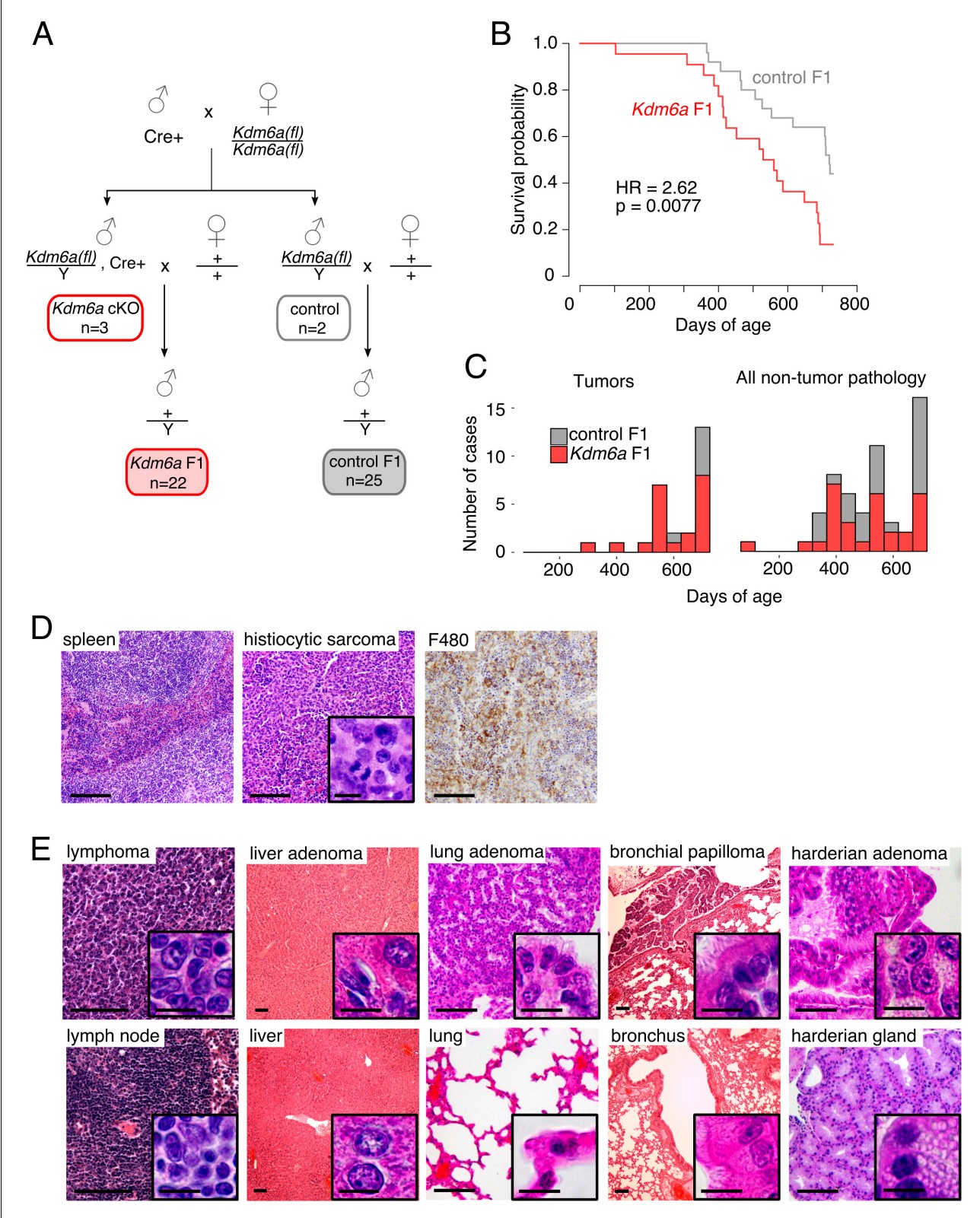

**Figure 1.** Reduced lifespan and increased tumor incidence in *Kdm6a* F1s. (**A**) Cross for *Kdm6a* F1 and control F1 mice. All *Kdm6a* cKO (n = 3) and control (n = 2) mice were littermates. (**B**) Survival curve for *Kdm6a* F1 and control F1 males. Hazard ratio (HR) and p-value calculated by a Cox proportional hazards model. (**C**) Raw counts of tumors (p=0.0071) and non-tumor phenotypes (p=0.69) in *Kdm6a* F1 vs. control F1 males at necropsy (p-values, one-sample test of proportions). (**D**) Left to right, hematoxylin and eosin (H&E) staining of normal spleen in control F1; H&E of histiocytic

*Figure 1 continued on next page*

*Figure 1 continued*

sarcoma in spleen of *Kdm6a* F1, showing diffuse infiltration of red pulp with nuclear pleomorphism and frequent mitotic figures (inset); immunohistochemistry of monocyte-lineage marker F4/80 in spleen histiocytic sarcoma. (**E**) H&E of representative tumors in *Kdm6a* F1s (top) and matched normal tissues from control F1s (bottom). Scale bars, 100 um (large images), 10 um (insets). See ***Figure 1—source data 1***.

DOI: https://doi.org/10.7554/eLife.39380.003

The following source data and figure supplements are available for figure 1:

**Source data 1.** Survival and phenotype of *Kdm6a* F1s.

DOI: https://doi.org/10.7554/eLife.39380.014

**Figure supplement 1.** Normal spermatogenesis in Utx cKO males.

DOI: https://doi.org/10.7554/eLife.39380.004

**Figure supplement 2.** Efficiency of *Ddx4-Cre* in the male germ line.

DOI: https://doi.org/10.7554/eLife.39380.005

**Figure supplement 3.** Survival of *Kdm6a* F1s from individual sires.

DOI: https://doi.org/10.7554/eLife.39380.006

**Figure supplement 4.** Survival of *Kdm6a* F1s grouped by presence or absence of the *Cre* transgene.

DOI: https://doi.org/10.7554/eLife.39380.007

**Figure supplement 5.** Contingency table for euthanasia vs.natural death in *Kdm6a* F1s and control F1s.

DOI: https://doi.org/10.7554/eLife.39380.008

**Figure supplement 6.** Utx F1 and control F1 weight and length.

DOI: https://doi.org/10.7554/eLife.39380.009

**Figure supplement 7.** Counts of gross and histopathological diagnoses at necropsy for *Kdm6a* F1s and control F1s.

DOI: https://doi.org/10.7554/eLife.39380.010

**Figure supplement 8.** Tumor rates in control and *Kdm6a* F1s and F2s broken down by individual sire.

DOI: https://doi.org/10.7554/eLife.39380.011

**Figure supplement 9.** Characterization of myeloid lineages in F1 bone marrow.

DOI: https://doi.org/10.7554/eLife.39380.012

**Figure supplement 10.** Validation of tumor types in Utx F1s.

DOI: https://doi.org/10.7554/eLife.39380.013

F1s that died between 12 and 18 months of age did not exhibit evidence of a unifying disease process. In contrast, *Kdm6a* F1s that died between 18 and 24 months of age exhibited an increased tumor burden compared to age-matched control F1s (***Figure 1C***, ***Figure 1—figure supplement 8***). The spectrum of tumors identified was similar to that observed in normally aging mice (***Haines et al., 2001***), but appeared earlier and at higher frequencies. The most common cancer type was histiocytic sarcoma, a blood tumor of the monocyte/macrophage lineage (***Figure 1D***; 6/22 vs. 1/25 mice, p=0.040, Fisher's Exact test); this tumor was found in *Kdm6a* F1s at a mean age of 624 ± 61 days, and in a single control F1 at 722 days. Flow cytometry of bone marrow from these mice revealed expanded populations of monocyte-lineage cells, consistent with histiocytic sarcoma. In addition, *Kdm6a* F1 mice not identified as having histiocytic sarcoma by histopathology also had a moderate increase in monocyte-lineage cell populations, indicating subtle skewing of hematopoietic lineages even in the absence of full-blown disease (***Figure 1—figure supplement 9***). *Kdm6a* F1 mice also developed a variety of other solid and blood tumors (***Figure 1E***, ***Figure 1—figure supplement 10***, ***Figure 2—source data 2***).

## Increased tumor susceptibility in Kdm6a F2 males

We then asked whether this effect could be transmitted to a second generation. We designed a breeding strategy in which wild type males were generated from male germ cells that had passed through two successive generations of *Kdm6a* conditional deletion ('*Kdm6a* F2'), or through one generation of *Kdm6a* deletion followed by one generation with an intact *Kdm6a* gene ('control F2') (***Figure 2A***). F2 males were followed under the same protocol as F1 males. We found that, like *Kdm6a* F1s, *Kdm6a* F2s exhibited reduced survival relative to the original control F1 cohort, whereas survival of control F2 males was more variable (***Figure 2B***, ***Figure 2—figure supplement 1***). Also like *Kdm6a* F1s, *Kdm6a* F2s had an increased tumor burden relative to the F1 control cohort (***Figure 2C and D***, ***Figure 2—figure supplement 2***). We did not find evidence for increased tumor

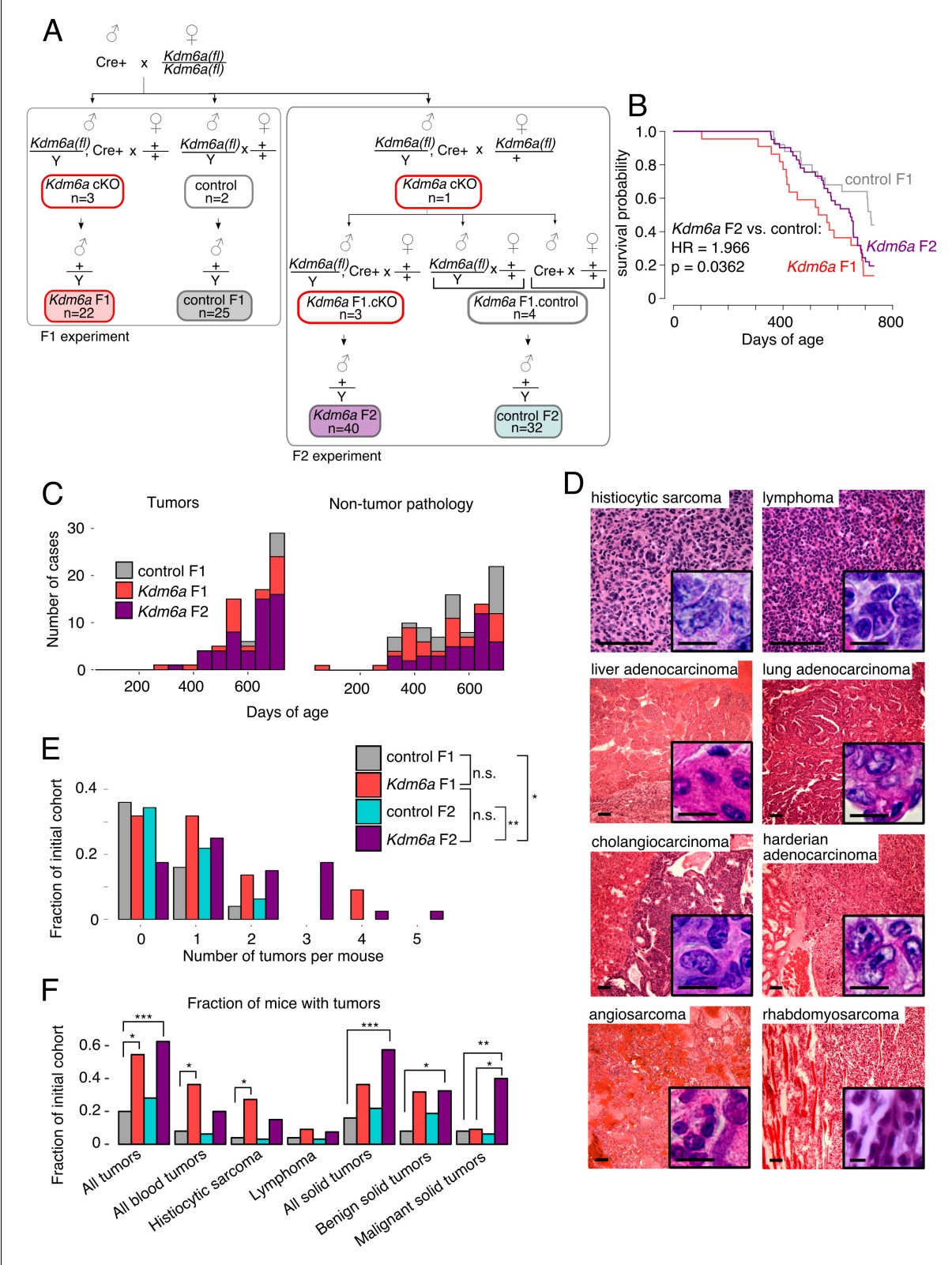

**Figure 2.** Reduced lifespan and increased tumor incidence in *Kdm6a* F2s. (**A**) Cross for *Kdm6a* F2s and control F2s. The *Kdm6a* cKO male used in this experiment was littermate to the 3 *Kdm6a* cKO and two control males used in the F1 experiment. Control F2s, combined progeny of *Cre*-only or only *Kdm6a(fl)*-only F1s. (**B**) Survival curve for *Kdm6a* F1s, control F1s, and *Kdm6a* F2s. Hazard ratio and p-value calculated by a Cox proportional hazards model. (**C**) Raw counts of tumors (p=3.45e-9) and non-tumor phenotypes (p=0.13) in control F1s, *Kdm6a* F1s, and *Kdm6a* F2s at necropsy (p-values, *Figure 2 continued on next page*

*Figure 2 continued*

*Kdm6a* F2s vs. control F1s, one-sample test of proportions). (D) H&E staining of representative tumors in *Kdm6a* F2s. Scale bar, 100 um (large images), 10 um (insets). (E) Tumor count per individual at necropsy. *p<0.05, **p<0.01, Fisher's exact test. (F) Fraction of mice with tumors. *p<0.05, **p<0.01, ***p<0.001, Fisher's exact test. See *Figure 2—source data 1*.

DOI: https://doi.org/10.7554/eLife.39380.015

The following source data and figure supplements are available for figure 2:

**Source data 1.** Survival and cancer phenotype of *Kdm6a* F2s.

DOI: https://doi.org/10.7554/eLife.39380.018

**Source data 2.** All tumors identified in F1 and F2 cohorts.

DOI: https://doi.org/10.7554/eLife.39380.019

**Figure supplement 1.** Survival of F2s from individual sires.

DOI: https://doi.org/10.7554/eLife.39380.016

**Figure supplement 2.** Counts of gross and histopathological diagnoses at necropsy for *Kdm6a* F1s, control F1s, and *Kdm6a* F2s.

DOI: https://doi.org/10.7554/eLife.39380.017

burden in control F2 males. We conclude that repeated loss of *Kdm6a* in the male germ line is required to maintain the intergenerational tumor susceptibility phenotype.

Notably, the tumor phenotype was more pronounced in *Kdm6a* F2s compared to *Kdm6a* F1s: *Kdm6a* F2s developed more tumors per mouse (overall tumor rate: 0.24 control, 0.95 *Kdm6a* F1, 1.30 *Kdm6a* F2; *Figure 2—source data 2*), and when present, tumors were more aggressive. Thirty-eight percent (15/40) of *Kdm6a* F2 mice had more than one independent tumor at death, compared to 23% (5/22) of *Kdm6a* F1 mice and 4% (1/25) of control F1 mice (*Figure 2E*). In addition, a higher fraction of *Kdm6a* F2 tumors were malignant (*Figure 2F*). We conclude that exposure of male germ cells to loss of *Kdm6a* across multiple generations confers a cumulative risk of tumor development on offspring. These findings imply that the molecular changes mediating this effect accumulate across generations, but can be reset when germline *Kdm6a* expression is restored.

## Altered epigenetic profiles in Kdm6a cKO male germ cells

We then turned our attention to the molecular mechanism by which loss of *Kdm6a* in the germ line might affect tumor susceptibility in the next generation. An advantage of our experimental strategy is that any epigenetic perturbation in germ cells is a consequence of a single defined genetic lesion, knockout of *Kdm6a*. We could therefore predict the nature of epigenetic changes in the *Kdm6a* cKO germ line based on the known molecular functions of the KDM6A protein. KDM6A is an H3K27me3 histone demethylase, and also plays a demethylase-independent role in promoting assembly of active enhancer regions (*Hong et al., 2007*; *Lan et al., 2007*; *Shpargel et al., 2012*; *Wang et al., 2017*). We first examined the effect of *Kdm6a* deletion on H3K27 methylation in male germ cells. We collected H3K27me3 ChIP-seq data from two biological replicates of epididymal sperm from *Kdm6a* cKO and littermate control males (*Figure 3—source data 2*, *Figure 3—figure supplement 1*). ChIP-seq data were strongly correlated between sperm replicates (*Figure 3—figure supplement 2*).

We examined H3K27me3 signal in 2-kilobase (kb) tiles throughout the genome. Genome-wide, we observed an increase in H3K27me3 signal in *Kdm6a* cKO sperm relative to control sperm after normalizing for library size, as expected for loss of an H3K27me3 demethylase (*Figure 3A*). We confirmed a global gain in H3K27me3 by Western blot (*Figure 3B*). However, this effect was not uniform throughout the genome. While H3K27me3 signal increased in the majority of tiles in *Kdm6a* cKO sperm, those tiles with the highest overall H3K27me3 signal exhibited a paradoxical loss of H3K27me3 in *Kdm6a* cKOs (*Figure 3C*, *Figure 3—figure supplement 3*). The result is an apparent flattening of the H3K27me3 profile: a decrease in H3K27me3 at regions with high signal, accompanied by an increase in H3K27me3 in adjacent regions (*Figure 3D–E*, *Figure 3—figure supplements 4–5*). This effect is compatible with several explanations. First, it may reflect genuine loss of signal in some regions accompanied by gain in adjacent regions. Second, widespread gain of H3K27me3 due to loss of KDM6A demethylase activity could result in the false appearance of signal loss at regions where H3K27me3 levels are actually unchanged. Finally, this effect may represent a more homogeneous signal at the population level due to increased variability between individual sperm. Allowing

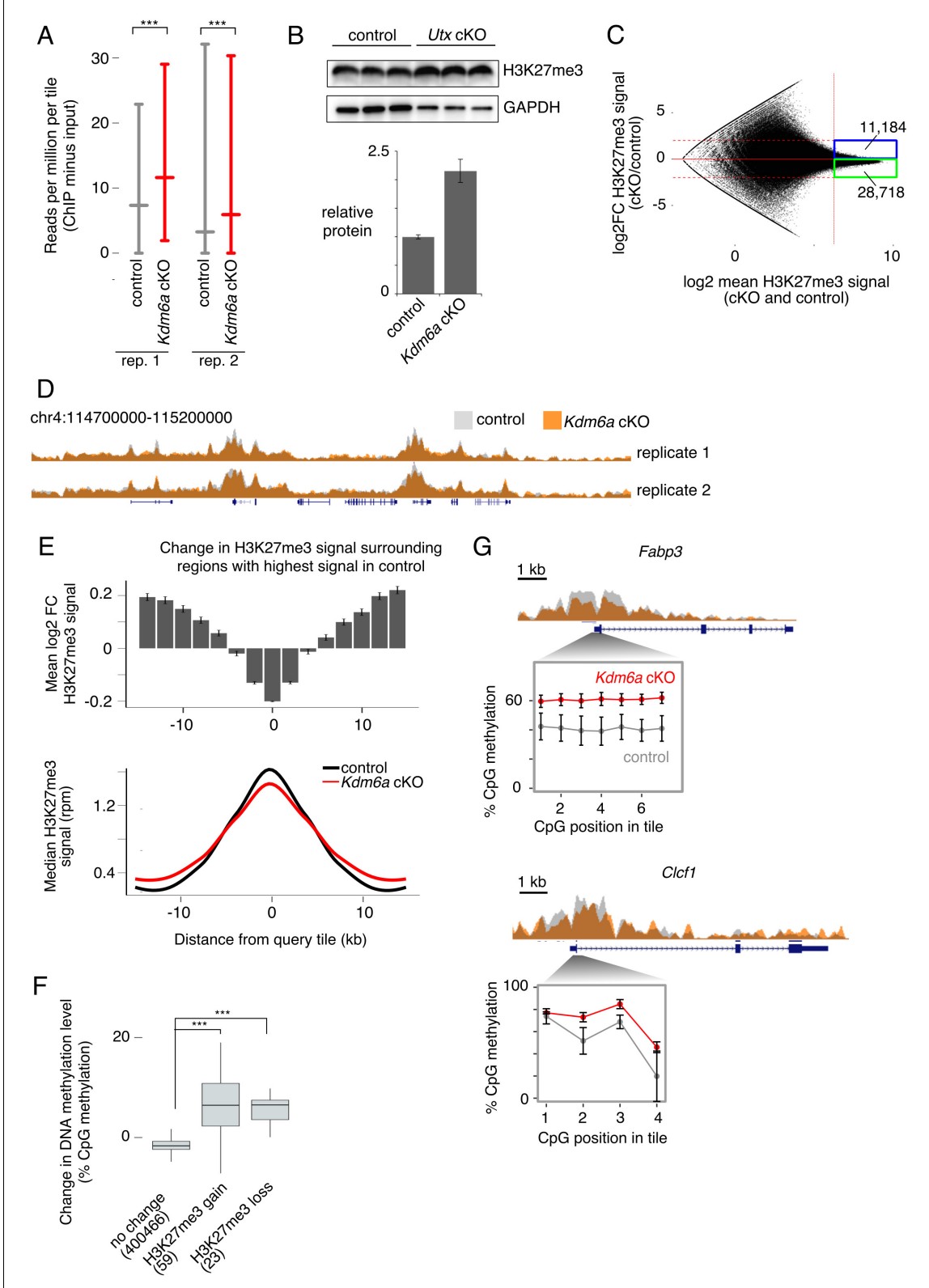

**Figure 3.** Redistribution of H3K27me3 in *Kdm6a* cKO germ cells. (**A**) Median and interquartile range (IQR) for H3K27me3 signal in 2 kb tiles for each of two sperm ChIP-seq replicates. ***p<2.2×10$^{-16}$, Mann-Whitney U test. (**B**) Western blot for H3K27me3 in germ cell-enriched testis samples from control and *Kdm6a* cKO mice. Bottom plot shows quantitation relative to GAPDH. Image is representative of two biological replicates. (**C**) MA plot of change in H3K27me3 signal vs. mean signal in *Kdm6a* cKO vs. control sperm, based on the mean of two biological replicates. Dashed horizontal lines, log2 fold

*Figure 3 continued on next page*

Figure 3 continued

change (log2FC) =±2. (**D**) Browser tracks of H3K27me3 signal in *Kdm6a* cKO and control sperm. (**E**) Top, mean log2FC in H3K27me3 signal for the 5% of tiles with greatest H3K27me3 signal in sperm and for surrounding tiles, based on mean values from two biological replicates. Error bars,±SE. Bottom, metagene of median H3K27me3 signal for the same set of tiles. (**F**) Change in DNA methylation level in *Kdm6a* cKO vs. control sperm for regions where log2FC H3K27me3 > 0.5 ('H3K27me3 gain'), log2FC H3K27me3 < −0.5 ('H3K27me3 loss'), or with no change in H3K27me3 (−0.5 < logFC < 0.5). Numbers of tiles in each category are shown. Horizontal bars, median; boxes, IQR. ***$p < 10^{-11}$, Mann-Whitney U test. (**G**) ChIP and RRBS data at two regions with altered H3K27me3 and DNA hypermethylation in sperm. Error bars, SEM of three replicates. See *Figure 3—source data 2*.
DOI: https://doi.org/10.7554/eLife.39380.020

The following source data and figure supplements are available for figure 3:

**Source data 1.** ChIP-seq libraries.
DOI: https://doi.org/10.7554/eLife.39380.029
**Source data 2.** H3K27me3 in *Kdm6a* cKO sperm.
DOI: https://doi.org/10.7554/eLife.39380.030
**Figure supplement 1.** Assay for purity of isolated epididymal sperm populations.
DOI: https://doi.org/10.7554/eLife.39380.021
**Figure supplement 2.** Correlations between individual datasets for genome-wide H3K27me3 ChIP-seq tiles.
DOI: https://doi.org/10.7554/eLife.39380.022
**Figure supplement 3.** MA plots of change in H3K27me3 signal vs.mean signal in *Kdm6a* cKO vs. control for individual sperm replicates.
DOI: https://doi.org/10.7554/eLife.39380.023
**Figure supplement 4.** Representative ChIP-seq browser tracks for control and *Kdm6a* cKO sperm.
DOI: https://doi.org/10.7554/eLife.39380.024
**Figure supplement 5.** Analysis of H3K27me3 changes in each sperm replicate.
DOI: https://doi.org/10.7554/eLife.39380.025
**Figure supplement 6.** Sample loci showing gain of DNA methylation in Utx cKO sperm.
DOI: https://doi.org/10.7554/eLife.39380.026
**Figure supplement 7.** Reanalysis of DNA methylation changes after exchanging data between replicates.
DOI: https://doi.org/10.7554/eLife.39380.027
**Figure supplement 8.** Characteristics of regions exhibiting reproducible changes in H3K27me3 in Utx cKO compared to control sperm.
DOI: https://doi.org/10.7554/eLife.39380.028

for each of these explanations, we conclude that loss of KDM6A increases H3K27me3 overall and alters the normal pattern of distribution of H3K27me3 during spermatogenesis.

Because we deleted *Kdm6a* early in spermatogenesis, we then considered the possibility that some epigenetic changes carried by *Kdm6a* cKO sperm might be indirect effects of early KDM6A loss. Deposition of H3K27 methylation has been associated with both gain and loss of cytosine DNA methylation, depending on the genomic and cellular context (*Brinkman et al., 2012*; *Neri et al., 2013*; *Viré et al., 2006*). DNA methylation is stable across long developmental time periods and is retained at high levels in sperm (*Monk et al., 1987*; *Smallwood et al., 2011*; *Smith et al., 2012*). We therefore asked if DNA methylation levels changed in regions of the genome where H3K27me3 was most perturbed in *Kdm6a* cKO relative to control sperm. We collected reduced representation bisulfite sequencing (RRBS) data from epididymal sperm of three control and three *Kdm6a* cKO males (*Figure 4—source data 2*). Overall levels of DNA methylation did not differ between control and cKO sperm (65% and 66% methylation, respectively). However, regions where H3K27me3 was altered, defined as those tiles with log2 fold change >0.5 or<−0.5 and false discovery rate <0.1 in both ChIP-seq replicates and which were not called as different in comparisons between the two control or two cKO datasets, were associated with increased DNA methylation (*Figure 3F and G*, *Figure 3—figure supplement 6*, *Figure 3—figure supplement 7*). Both increased and decreased H3K27me3 were associated with a gain in DNA methylation, possibly due to secondary alterations in histone methylation after establishment of an initial change in DNA methylation. These regions were enriched near gene bodies (p=$9.898 \times 10^{-6}$ for H3K27me3 gain and p=$5.892 \times 10^{-4}$ for H3K27me3 loss, Fisher's exact test), and regions of H3K27me3 loss were also weakly enriched at transcription start sites (p=0.01368, Fisher's exact test). Genes exhibiting loss of H3K27me3 and gain of DNA methylation were enriched for functions such as 'negative regulation of myeloid dendritic cell activation' and 'positive regulation of immune effector process' (*Figure 3—figure supplement 8*). Together, our results indicate that deletion of *Kdm6a* early in spermatogenesis induces redistribution

of H3K27me3, and that regions strongly affected by H3K27me3 redistribution gain DNA methylation in mature sperm.

## Differential DNA methylation persists from Kdm6a cKO sperm to Kdm6a F1 soma

We then asked if the changes in DNA methylation evident in *Kdm6a* cKO sperm could also be detected in somatic tissues of aging *Kdm6a* F1 adults. We collected RRBS data from bone marrow of *Kdm6a* F1 and control F1 males (*Figure 4—source data 2*), and compared it to the RRBS data from *Kdm6a* cKO and control sperm. We identified differentially methylated regions (DMRs: 100 bp tiles with false discovery rate <0.05) in *Kdm6a* cKO vs. control sperm and in *Kdm6a* F1 vs. control F1 bone marrow (*Figure 4A*). To avoid the confounding effect of disease on DNA methylation, we excluded F1 mice with any histopathological abnormality in the blood lineage. DMRs in both *Kdm6a* cKO sperm and *Kdm6a* F1 bone marrow were more likely to be hypermethylated than hypomethylated relative to their respective controls (4725 hypermethylated vs. 323 hypomethylated DMRs in sperm and 3156 hypermethylated vs. 1122 hypomethylated DMRs in bone marrow). Two hundred and ninety-nine regions were differentially methylated in both *Kdm6a* cKO sperm and *Kdm6a* F1 bone marrow, significantly more than expected by chance (57 regions expected, p=4.22e-121, hypergeometric test) (*Figure 4B*). Considering all 299 shared DMRs, there was a positive correlation between the magnitude of DNA methylation change in sperm and in F1 bone marrow (R = 0.17, p=0.0026) (*Figure 4C*). Two hundred and twenty-six individual DMRs (76%) were positively correlated between sperm and F1 bone marrow, including 207 (69%) hypermethylated and 19 (6%) hypomethylated regions (*Figure 4—source data 3*, *Figure 4—source data 2*). Given the overall hypermethylation of DMRs in both *Kdm6a* cKO sperm and *Kdm6a* F1 bone marrow, we focused our attention on the 207 hypermethylated regions. We considered these positively-correlated hypermethylated DMRs as candidates for direct inheritance of DNA methylation state from the paternal germ line, and refer to them as 'persistent' DMRs. We validated our RRBS findings using pyrosequencing in *Kdm6a* cKO sperm and *Kdm6a* F1 bone marrow, and confirmed hypermethylation at 12 of 13 tested DMRs in at least one tissue and at seven of 13 DMRs in both tissues (*Figure 4—figure supplement 1*).

## Persistent Kdm6a DMRs overlap enhancers associated with tumorigenesis

We then asked what genomic and regulatory features were associated with persistent DMRs. We found that persistent DMRs were distributed throughout the genome (*Figure 4D*) and frequently overlapped the regions of greatest H3K27me3 change in *Kdm6a* cKO sperm (*Figures 3F* and *4E*). In contrast, there was no association between persistent DMRs and various other features, including CpG islands, imprinted regions, and transcription start sites (TSS) (*Figure 4F*, *Figure 4—figure supplement 2*). Although repetitive elements such as retrotransposons can be resistant to DNA methylation reprogramming in the germ line (*Guibert et al., 2012*), persistent DMRs were not more likely to overlap repetitive elements compared to the complete set of genomic regions covered by our RRBS data (*Figure 4G*). We conclude that the location of persistent DMRs is strongly associated with regions of altered H3K27me3 in sperm, implying that loss of *Kdm6a* in the male germ line sensitizes these regions to DNA hypermethylation. Some of these sensitive regions may retain their methylation state during somatic development in the next generation.

We next asked whether persistent DMRs might be functionally important to the tumor susceptibility phenotype observed in *Kdm6a* F1s. We examined the proximity of persistent DMRs to enhancer regions in whole bone marrow and in sorted bone marrow macrophages (mouse ENCODE project) (*Yue et al., 2014*) and in round spermatids, the last stage of spermatogenesis at which there is active transcription (our data, *Figure 3—source data 2*; *Figure 4—figure supplement 3*). We found that persistent DMRs were close to or overlapping both poised (marked by H3K4me1) and active (marked by both H3K4me1 and H3K27ac) enhancer regions in all three of these tissues or cell types (*Figure 4H*, *Figure 4—figure supplement 2*). We then used GREAT (Genomic Regions Enrichment of Annotations Tool) to identify enriched phenotypes, defined by the Mouse Genome Informatics (MGI) phenotype ontology, associated with the set of 207 persistent DMRs (*Blake et al., 2009*; *McLean et al., 2010*). The top ten most strongly enriched mouse phenotypes were all related

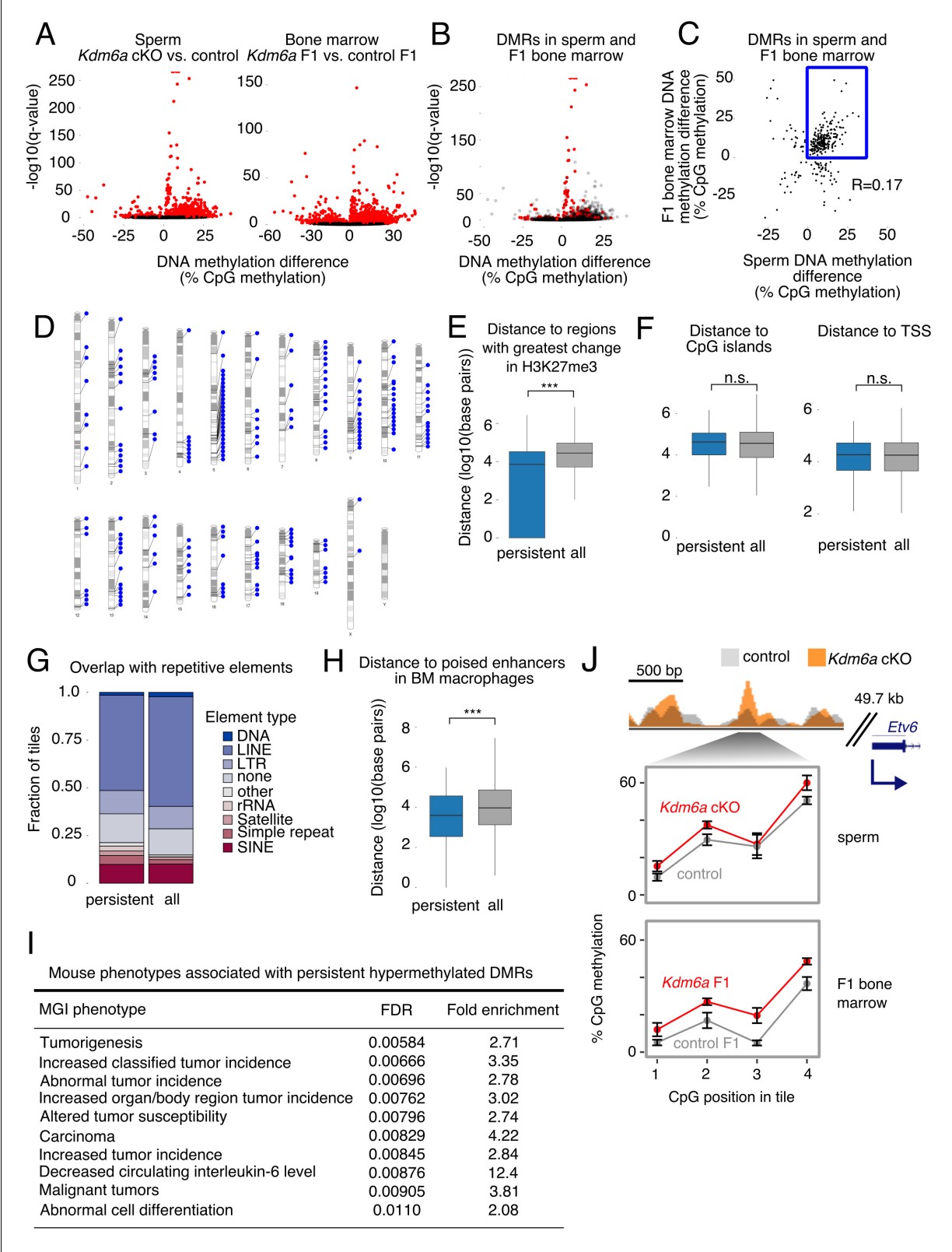

**Figure 4.** Persistent DMRs are associated with altered H3K27me3 and enhancer regions. (**A**) Left, differentially methylated regions (DMRs) in sperm. Right, DMRs in F1 bone marrow. Red, false discovery rate (FDR) < 0.05. (**B**) Sperm volcano plot from (**A**); DMRs with FDR < 0.05 in both sperm and F1 bone marrow are in red. (**C**) Magnitude of DNA methylation difference (*Kdm6a* cKO vs. control or *Kdm6a* F1 vs. control F1) for the 299 DMRs shared between sperm and F1 bone marrow. Box, persistent DMRs. (**D**) Distribution of persistent DMRs in the mouse genome. (**E**) Distance from persistent

*Figure 4 continued on next page*

*Figure 4 continued*

DMRs to the 25% of regions with greatest change in H3K27me3 in sperm. 'All' refers to the complete set of tiles covered by RRBS. ***p<0.001, Mann-Whitney U test. (F) Left, distance to CpG islands. Right, distance to transcription start sites (TSS). (G) Fraction of DMRs overlapping repetitive elements. (H) Distance to poised enhancers in sorted bone marrow macrophages. ***p<0.001, Mann-Whitney U test. (I) Top 10 mouse phenotypes associated with persistent DMRs. (J) Representative persistent DMR in the enhancer of a cancer-associated gene (*Etv6*). Error bars, SEM of three replicates. See *Figure 4—source data 1*.

DOI: https://doi.org/10.7554/eLife.39380.031

The following source data and figure supplements are available for figure 4:

**Source data 1.** DNA methylation in *Kdm6a* cKO sperm and *Kdm6a* F1 bone marrow.
DOI: https://doi.org/10.7554/eLife.39380.037
**Source data 2.** RRBS libraries.
DOI: https://doi.org/10.7554/eLife.39380.038
**Source data 3.** DMRs shared between *Kdm6a* cKO sperm and *Kdm6a* F1 bone marrow.
DOI: https://doi.org/10.7554/eLife.39380.039
**Source data 4.** Genes within 1 kilobase of persistent DMRs.
DOI: https://doi.org/10.7554/eLife.39380.040
**Figure supplement 1.** Representative pyrosequencing data at three persistent DMRs, including two tumor-associated enhancers (*Foxa2* and *Lmo2*) and one promoter (*Lama3*).
DOI: https://doi.org/10.7554/eLife.39380.032
**Figure supplement 2.** Distance relationships between persistent DMRs and various genomic features.
DOI: https://doi.org/10.7554/eLife.39380.033
**Figure supplement 3.** Sorting of round spermatids by flow cytometry.
DOI: https://doi.org/10.7554/eLife.39380.034
**Figure supplement 4.** Additional examples of DNA methylation gains in sperm and F1 bone marrow.
DOI: https://doi.org/10.7554/eLife.39380.035
**Figure supplement 5.** Additional examples of DNA methylation gains in sperm and F1 bone marrow in enhancers associated with tumorigenesis.
DOI: https://doi.org/10.7554/eLife.39380.036

to tumorigenesis, including 'increased classified tumor incidence', 'altered tumor susceptibility', and 'malignant tumors' (*Figure 4I–J*, *Figure 4—figure supplements 4–5*). We conclude that *Kdm6a*-dependent hypermethylated persistent DMRs affect enhancer regions relevant to tumorigenesis in mice. We note that the edges of a ChIP-seq peak do not represent precise boundaries for functional enhancer regions, meaning that DMRs that are close to but not directly overlapping enhancers in our analysis may still affect their function, for example by altering local transcription factor binding affinities or long-range chromatin interactions (*Tiwari et al., 2008*; *Yin et al., 2017*; *Onuchic et al., 2018*).

To test the hypothesis that methylation changes persisted from sperm through the early embryo to adult tissue, we also evaluated DNA methylation changes in spleens of five control F1 and three *Kdm6a* F1 mice, and in liver tumors from two control F1s and two *Kdm6a* F1s. Of the 207 persistent DMRs detected in bone marrow, 140 (67%, OR 87.32, p<2.2e-16) were also found in liver tumors, and 68 (35%, OR 408.65, p<2.2e-16) were also found in spleen, and the magnitudes of DNA methylation changes were positively correlated: R = 0.232 (liver) and R = 0.786 (spleen). The similarity of methylation changes across different tissues supports the model that these changes were present in the early embryo and persisted during lineage commitment and organ differentiation.

## Persistent Kdm6a DMRs can alter transcription factor binding at enhancers

One effect of DNA methylation at enhancers is to modulate the binding affinities of recruited transcription factors (TFs), thereby altering downstream regulatory circuitry (*Yin et al., 2017*). We therefore investigated the possibility that the set of persistent DMRs contains methylation-sensitive TF binding sites that can impact expression of nearby genes. We used AME (Analysis of Motif Enrichment) (*McLeay and Bailey, 2010*) to find enriched TF binding motifs in the set of persistent DMRs. We detected enrichment of binding sites corresponding to the ETS transcription factors ELK1, ELK4, and GABPA (*Figure 5A*). DNA methylation reduces the affinity of all three of these factors for their

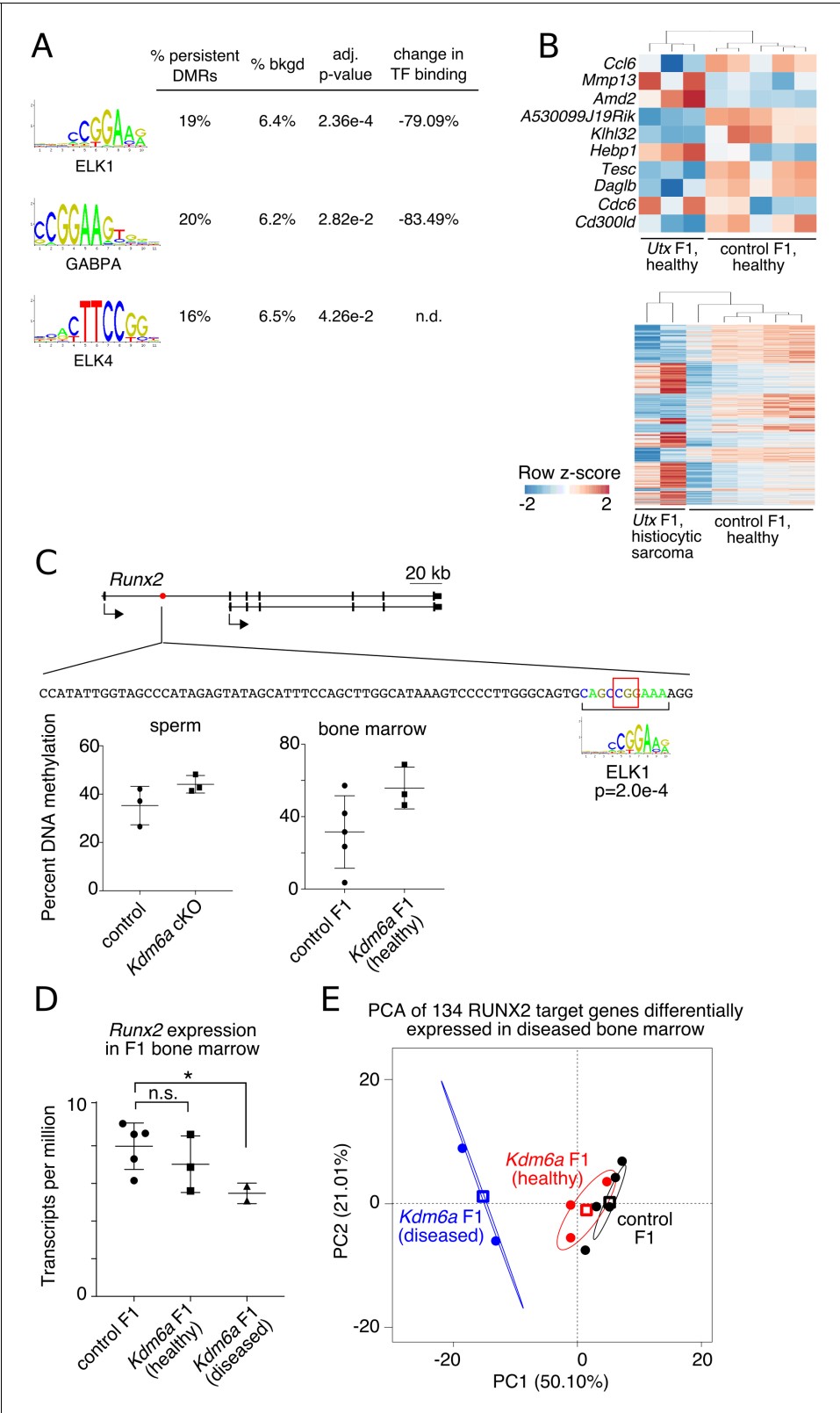

**Figure 5.** Persistent DMRs affect F1 bone marrow expression profiles. (**A**) Transcription factor (TF) binding sites enriched in persistent DMRs. 'Adjusted p-value': Bonferroni-corrected AME p-value. '% persistent DMRs', '% background': percentage of tiles containing the TF binding site. 'Change in binding', relative enrichment of mCpGs in bisulfite-SELEX data from *Yin et al. (2017)*. (**B**) Genes differentially expressed in healthy *Kdm6a* F1 (top) or diseased *Kdm6a* F1 (bottom) vs. control F1 bone marrow. (**C**) Top, gene model of *Runx2* with location of a persistent DMR in the first intron (red circle). *Figure 5 continued on next page*

*Figure 5 continued*

Middle, sequence of the DMR including ELK1 binding site and affected CpG (red box). Bottom, RRBS DNA methylation levels at the boxed CpG in sperm and F1 bone marrow. (**D**) Expression of *Runx2* in F1 bone marrow. *p<0.05, Welch's t-test. (**E**) Principal component analysis of 134 differentially expressed RUNX2 target genes. Circles, individual samples; open squares, centroid; ellipses, 95% confidence interval. See *Figure 5—source data 1*.

DOI: https://doi.org/10.7554/eLife.39380.041

The following source data is available for figure 5:

**Source data 1.** *Runx2* expression and regulation in *Kdm6a* F1 bone marrow.

DOI: https://doi.org/10.7554/eLife.39380.042

binding sites (*Yin et al., 2017*), implying that persistent hypermethylation at these sites can impact expression of downstream genes in F1 somatic tissue.

To evaluate this possibility, we collected RNA-seq data from bone marrow of healthy *Kdm6a* F1s (n = 3), *Kdm6a* F1s with abnormal histiocytic proliferation or sarcoma (n = 2), and healthy control F1s (n = 5), and looked for transcriptional signatures consistent with altered regulation by ELK1, ELK4, or GABPA. We called differentially expressed genes (adjusted p-value<0.05) for healthy *Kdm6a* F1s vs. control F1s and for diseased *Kdm6a* F1s vs. control F1s separately (*Figure 5B*). In keeping with our prediction, four of ten differentially expressed genes in healthy *Kdm6a* F1 bone marrow and 134 of 1404 differentially expressed genes in diseased *Kdm6a* F1 bone marrow were targets of the hematopoiesis-associated transcription factor RUNX2, a direct target of ELK1 (p=0.00492 and p=0.00102 for healthy and diseased *Kdm6a* F1 bone marrow, respectively, Fisher's exact test) (*Matys et al., 2003*; *Zhang et al., 2009*). An ELK1 binding site in the first intron of *Runx2* falls within a persistent hypermethylated DMR and exhibits increased DNA methylation in both *Kdm6a* cKO sperm and *Kdm6a* F1 bone marrow (*Figure 5C*). Expression of *Runx2* itself was decreased in diseased *Kdm6a* F1 compared to control F1 bone marrow (*Figure 5D*). Principal component analysis of expression data for the 134 differentially expressed RUNX2 target genes placed healthy *Kdm6a* F1 between diseased *Kdm6a* F1 and control F1 bone marrow, revealing potential underlying similarities in regulation of the *Runx2* transcriptional network among *Kdm6a* F1 samples (*Figure 5E*). Although the observed effect was small and should be confirmed in additional tissues, this result implies that altered regulation of transcriptional networks downstream of DNA methylation-sensitive transcription factors could result from persistent DNA hypermethylation transmitted from the *Kdm6a* cKO germ line to F1 somatic tissue.

## Discussion

### A model for epigenetic inheritance of cancer susceptibility

We propose a model (*Figure 6*) wherein loss of *Kdm6a* results in extensive redistribution of the H3K27me3 mark during male germ cell development. Retention of H3K27me3 at regions where it would ordinarily be turned over leaves some of these loci vulnerable to DNA methylation, leading to hypermethylation in sperm. Early in embryogenesis, when most DNA methylation is removed from the paternal genome, some of these hypermethylated regions may resist reprogramming, such that methylation persists in somatic tissue during development; alternatively, hypermethylation may be lost at these loci during reprogramming and reestablished later in development following transmission through an epigenetic intermediate. When these regions coincide with functional enhancers, the altered epigenetic state inherited from the paternal gamete can have transcriptional consequences.

Importantly, because each sperm carries only one copy of a given locus, the relatively modest shift we observe in DNA methylation levels must reflect variability among individual sperm. Such variability is consistent with the heterogeneous tumor profiles and other pathological phenotypes seen in our F1 population. Similarly, in F1 somatic tissues, we propose that the effects of DNA methylation on downstream gene regulation manifest as a shift in the probability of transcription factor binding, resulting in subtle changes to transcriptional networks that impact tissue function only in the context of stressors, or cumulatively over time.

Several key questions remain to be answered. First, it will be important to define the role of *Uty*, the Y-linked homolog of *Kdm6a* that lacks histone demethylase activity, in this phenomenon

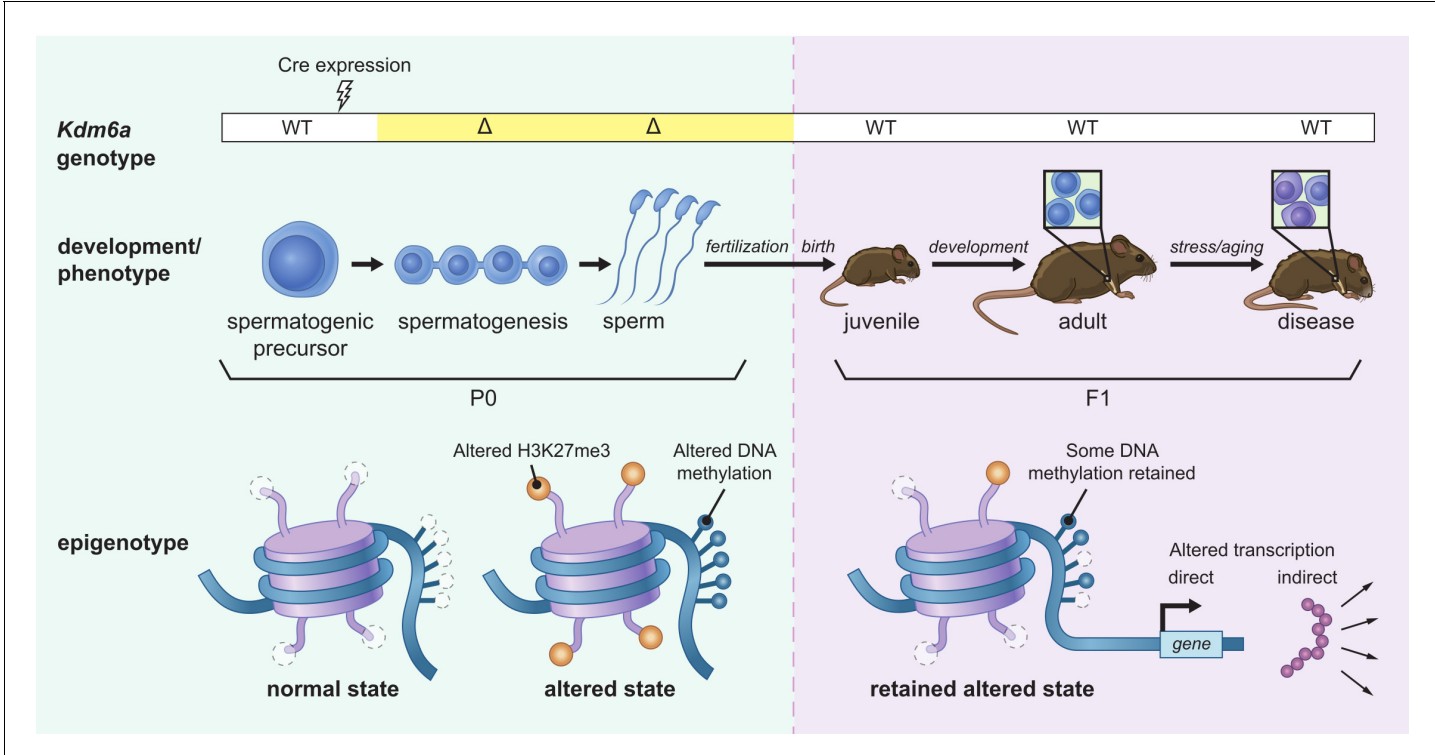

**Figure 6.** Model for intergenerational epigenetic inheritance following deletion of *Kdm6a* in the male germ line. *Kdm6a* excision occurs in early spermatogenic precursors, resulting in genome-wide changes in H3K27me3 distribution. Altered H3K27me3 distribution biases nearby regions toward gain of DNA methylation. Both H3K27me3 and DNA methylation changes are retained in mature sperm. At fertilization, H3K27me3 and most DNA methylation changes are reset, but some DNA methylation gains persist. DNA methylation gains influence expression of nearby genes during development in genetically wild type F1 offspring. Effects on phenotype occur when downstream gene regulatory circuits are subjected to environmental or aging-associated stress.

DOI: https://doi.org/10.7554/eLife.39380.043

(*Hong et al., 2007*). Second, while we have focused on adult cancer phenotypes, it is possible that additional developmental phenotypes are also affected. We did observe some developmental anomalies in adult *Kdm6a* progeny, including ectopic tissue rests, tail kinks, scoliosis, and a thyroglossal duct cyst. Third, the extent to which premature appearance of age-associated tumors might reflect a more generalized premature aging phenotype should be examined in more depth.

It will be critical to dissect the underlying molecular mechanism in more detail. While we suggest that DNA methylation changes induced during spermatogenesis persist during reprogramming in the early embryo, we have not yet directly demonstrated that this is the case. It is also possible that DNA hypermethylation is lost during reprogramming, but that epigenetic information is transmitted through an alternative chromatin mark or RNA intermediate to reestablish DNA hypermethylation later in development. Assessment of DNA methylation in early *Kdm6a* F1 embryos will help to resolve this question.

The relationship between *Kdm6a* loss, redistribution of H3K27me3, and gain of DNA methylation also remains to be defined. The simplest explanation for our data is that dysregulation of H3K27me3 leads to DNA hypermethylation at vulnerable loci, but it is also possible that KDM6A acts through an independent mechanism to regulate DNA methylation. Determination of the stage of spermatogenesis (proliferating spermatogonia, meiotic spermatocytes, or haploid spermatids) at which the observed changes in H3K27me3 and DNA methylation first appear will help to delineate the relationship between the different epigenetic consequences of *Kdm6a* loss.

Intriguingly, a recent study of H3K27me3 in mouse embryonic stem cells (mESCs) grown in 2i compared to serum-containing media described flattening of H3K27me3 signal very similar to the effect we observed in *Kdm6a* cKO sperm (*van Mierlo et al., 2019*). In 2i mESCs, H3K27me3

flattening was also associated with altered DNA methylation. A closer examination of the relationship between the phenomenon we observe in sperm and that reported in 2i mESCs may shed light on the mechanisms underlying both phenomena.

Finally, a critical experiment will be to examine the sperm of *Kdm6a* F1s to test the prediction that changes in DNA methylation at persistent DMRs are amplified in the second generation of gametes. At least two persistent DMRs are located in or near genes encoding components of the DNA methylation machinery (*Dnmt3a* and *Tdg*), raising the possibility that DNA hypermethylation at these sites in sperm amplifies the changes in DNA methylation in offspring. *Dnmt3a* is frequently mutated in hematological tumors and has been defined as an important tumor suppressor (*Yang et al., 2015*).

We restricted our study to progeny of a single male founder in order to limit the amount of genetic variation in the experiment and thereby reduce the potential contribution of genetic heterogeneity, given a moderate number of experimental animals (~100 F1s and F2s total). However, our findings should be tested in a larger study including several founder males. Likewise, a larger study would allow recovery of more diseased samples for transcriptional analysis. It will also be critical to exclude the possibility that loss of KDM6A in the male germ line leads to increased DNA damage and accumulation of genomic mutations that could contribute to a tumor phenotype in the next generation. Since increased DNA damage during spermatogenesis frequently leads to meiotic arrest and impaired fertility, the normal spermatogenesis and fertility of *Kdm6a* cKO mice argue against a strong mutator phenotype (*Hunt and Hassold, 2002*). However, a more subtle effect should be ruled out by sequencing of genomic DNA in multiple F1 progeny and careful assessment of mutation rates.

## Implications for human disease

Virtually nothing is known about the contribution of epigenetic perturbations in the male germ line to human disease susceptibility. Specifically, while increased attention is being paid to the possible impacts of diet and environmental exposure on male fertility and epigenetic inheritance (*Anway et al., 2005*; *Carone et al., 2010*; *Kaati et al., 2002*; *Ly et al., 2017*), the role of mutations that arise in the male germ line but are not transmitted to the next generation is entirely unknown and unexplored. Spermatogenic stem cells continue to divide and to accumulate de novo mutations throughout a man's lifetime. De novo germline mutations linked to advanced paternal age have been implicated in the pathogenesis of autism and schizophrenia; in these cases, the causative mutations arise in the germ line and are inherited by the affected progeny (*Awadalla et al., 2010*; *de Kluiver et al., 2017*; *Girard et al., 2011*; *Iossifov et al., 2014*; *Nybo Andersen and Urhoj, 2017*). Our results imply that de novo mutations in the male germ line in genes such as *Kdm6a* may have phenotypic consequences for progeny, even when they are not inherited. Intergenerational paternal effects on development have also been reported for heterozygous autosomal mutations in genes encoding chromatin regulators in the mouse (*Chong et al., 2007*), suggesting that the effects of non-inherited paternal germline mutations do not depend on complete loss of gene function in the germ cells. Interestingly, a paternal age effect has been reported for ALL, a tumor shown to be sensitive to epigenetic regulation by KDM6A, but increased rates of inherited de novo mutations have not yet been demonstrated for ALL patients (*Sergentanis et al., 2015*).

Many patients with cancer are now being treated with drugs that target epigenetic regulators. If these drugs alter the epigenetic state of germ cells, these treatment protocols could have long-term consequences for offspring of fertile patients. Based on the findings reported here and previously (*Carone et al., 2010*; *Chong et al., 2007*; *Kaati et al., 2002*; *Pembrey et al., 2006*; *Siklenka et al., 2015*), we suggest that paternal epigenetic state should be evaluated as an important risk factor in human disease susceptibility.

# Materials and methods

**Key resources table**

| Reagent type (species) or resource | Designation | Source or reference | Identifiers | Additional information |
|---|---|---|---|---|

*Continued on next page*

*Continued*

| Reagent type (species) or resource | Designation | Source or reference | Identifiers | Additional information |
|---|---|---|---|---|
| Gene (*Mus musculus*) | *Kdm6a* | NA | MGI:1095419 | Also called *Utx* |
| Genetic reagent (*Mus musculus*) | *Ddx4-Cre* | **Hu et al., 2013** | B6-*Ddx4*$^{tm1.1(cre/mOrange)Dcp}$ RRID:MGI:5554603 | Also called *Mvh-Cre* |
| Genetic reagent (Mus musculus) | *Kdm6a(fl)* | **Welstead et al., 2012** | B6;129S4-Kdm6atm1 c(EUCOMM)Jae/J RRID:IMSR_JAX:021926) | |
| Antibody | Mouse monoclonal anti-H3K27me3 | Abcam | ab6002 RRID:AB_305237 | 1:1000 |
| Antibody | Rabbit polyclonal anti-H3K27me3 | Millipore Sigma | 07–449 RRID:AB_310624 | 1:1000 |
| Antibody | Rabbit polyclonal anti-H3K4me1 | Abcam | ab8895 RRID:AB_306847 | 1:1000 |
| Antibody | Rabbit polyclonal anti-H3K27ac | Abcam | ab4729 RRID:AB_2118291 | 1:1000 |
| Antibody | Mouse monoclonal anti-Gapdh | Santa Cruz Biotechnology | sc-32233 RRID:AB_627679 | 1:1000 |
| Antibody | Rat monoclonal anti-F4/80 | Serotec | MCA497GA RRID:AB_323806 | clone CI:A3-1; 1:5000 |
| Antibody | Rabbit monoclonal anti-VEGF-A | Abcam | ab52917 RRID:AB_883427 | clone EP1176Y; 1:100 |
| Antibody | Rabbit monoclonal anti-ERG | Abcam | ab133264 RRID:AB_11156852 | 1:250 |
| Antibody | Rabbit monoclonal anti-TTF-1 | Abcam | ab76013 RRID:AB_1310784 | 1:250 |
| Antibody | Rabbit polyclonal anti-GS | Abcam | ab73593 RRID:AB_2247588 | 1:1000 |
| Antibody | Mouse monoclonal anti-CD20 | Dako | M0755 RRID:AB_2282030 | clone L26; 1:500 |
| Antibody | Rabbit polyclonal anti-CD3 | Dako | A0452 RRID:AB_2335677 | clone F7.2.38; 1:250 |
| Sequence-based reagent | RT-qPCR primer, *Actb*-F | this paper | AGAAGGACTCCTATGTGGGTGA | |
| Sequence-based reagent | RT-qPCR primer, *Actb*-R | this paper | CATGATCTGGGTCATCTTTTCA | |
| Sequence-based reagent | RT-qPCR primer, *Sycp2*-F | this paper | AGTCTGAGCTGATGTTATCATA | |
| Sequence-based reagent | RT-qPCR primer, *Sycp2*-R | this paper | GAAGCAGAAGTAGAAGAGGC | |
| Sequence-based reagent | RT-qPCR primer, *Prm2*-F | this paper | GCTGCTCTCGTAAGAGGCTACA | |
| Sequence-based reagent | RT-qPCR primer, *Prm2*-R | this paper | AGTGATGGTGCCTCCTACATTT | |
| Sequence-based reagent | RT-qPCR primer, *Aqp8*-F | this paper | GGATGTCTATCGGTCATTGAG | |
| Sequence-based reagent | RT-qPCR primer, *Aqp8*-R | this paper | GAATTAGCAGCATGGTCTTGA | |
| Sequence-based reagent | RT-qPCR primer, *Lin28a*-F | this paper | TGGTGTGTTCTGTATTGGGAGT | |
| Sequence-based reagent | RT-qPCR primer, *Lin28a*-R | this paper | AGTTGTAGCACCTGTCTCCTTT | |
| Commercial assay or kit | Zymo ChIP Clean and Concentrator kit | Zymo Research | D5201 | |

*Continued on next page*

*Continued*

| Reagent type (species) or resource | Designation | Source or reference | Identifiers | Additional information |
|---|---|---|---|---|
| Commercial assay or kit | Accel-NGS 2S plus DNA library kit | Swift Biosciences | 21024 | |
| Commercial assay or kit | DNEasy Blood andTissue kit | Qiagen | 69504 | |
| Commercial assay or kit | Ovation RRBS Methyl-Seq System | NuGen | 0353 | |
| Commercial assay or kit | RNEasy Plus Mini kit | Qiagen | 74134 | |
| Commercial assay or kit | TruSeq RNA library prep kit | Illumina | RS-122–2001 | |
| Software, algorithm | R | R Core Team | RRID:SCR_001905 | https://http://www.R-project.org/ |
| Software, algorithm | Fastx toolkit v0.0.14 | http://hannonlab.cshl.edu/fastx_toolkit/commandline.html | RRID:SCR_005534 | |
| Software, algorithm | MACS v1.4 | *Zhang et al., 2008* | RRID:SCR_013291 | |
| Software, algorithm | MACS v2.1 | *Zhang et al., 2008* | RRID:SCR_013291 | |
| Software, algorithm | bowtie v1.2 | *Langmead et al., 2009* | RRID:SCR_005476 | |
| Software, algorithm | bowtie v2.0 | *Langmead and Salzberg, 2012* | RRID:SCR_016368 | |
| Software, algorithm | trim-galore v0.4.2 | https://www.bioinformatics.babraham.ac.uk/projects/trim_galore/ | RRID:SCR_011847 | |
| Software, algorithm | bismark v0.16.3 | *Krueger and Andrews, 2011* | RRID:SCR_005604 | |
| Software, algorithm | phenogram | http://visualization.ritchielab.psu.edu/phenograms/document | | |
| Software, algorithm | DESeq2 (R package) | *Love et al., 2014* | RRID:SCR_015687 | |
| Software, algorithm | kallisto v0.43.0 | *Bray et al., 2016* | RRID:SCR_016582 | |
| Software, algorithm | AME | *McLeay and Bailey, 2010* | RRID:SCR_001783 | |
| Software, algorithm | methylKit (R package) | *Akalin et al., 2012* | RRID:SCR_005177 | |
| Software, algorithm | rms (R package) | https://cran.r-project.org/web/packages/rms/index.html | | |
| Software, algorithm | survival (R package) | https://CRAN.R-project.org/package=survival | | |
| Software, algorithm | FactoMineR (R package) | *Le et al., 2008* | RRID:SCR_014602 | |

## Experimental design

This experiment was designed to test the hypothesis that epigenetic changes in the germ line resulting from loss of KDM6A could induce gross phenotypic or survival changes in genetically wild type offspring. The F1 experiment was 80% powered to detect a survival hazard ratio of 2.5 and 90% powered to detect a 2.5-fold change in phenotype incidence. The F2 experiment was 90% powered to detect a survival hazard ratio of 2.5 and 95% powered to detect a 2.5-fold change in phenotype incidence. Type I error rate (alpha) was 5% for all power calculations.

## Statistical analysis

Survival hazard ratios were calculated using a Cox proportional hazards model. Fisher's exact test was used to compare proportions. Welch's t-test was used to compare continuous, normally-distributed variables. A Mann-Whitney U test was used for continuous variables when a normal distribution could not be assumed.

## Mouse breeding and husbandry

All mice were maintained at the Whitehead Institute animal facility. Mice were kept under standard conditions and all experiments were conducted in compliance with the Animal Welfare act and approved by the Animal Care and Use Committee at the Massachusetts Institute of Technology. *Kdm6a* cKO, control, and all F1 and F2 mice were generated with breeding schemes described in the main text using *Ddx4-Cre* (B6-*Ddx4*$^{tm1.1(cre/mOrange)Dcp}$) (*Hu et al., 2013*) and *Kdm6a(fl)* (B6;129S4-*Kdm6a*$^{tm1c(EUCOMM)Jae}$/J) (*Welstead et al., 2012*) alleles. Experiments were carried out on a mixed C57BL/6, 129S4 background. We controlled for background effects by generating all experimental mice from a single founder male, generating experimental F1s and F2s from littermate *Kdm6a* cKO and control males, and by removing loci containing known B6/129 variants from downstream analysis of DMRs. To generate F1 and F2 mice, single males were continuously co-housed with single C57BL/6 females, and litters were weaned at three weeks of age. All control and experimental mice were housed with littermates in adjacent cages on the same rack and subjected to identical handling protocols.

## Necropsy and histopathology

F1 and F2 mice were checked daily for morbidity and mortality beginning at 6 months of age. Mice that died spontaneously were recovered within 24 hr to avoid autolysis and underwent a full necropsy. Mice that were independently identified by the MIT veterinary staff as requiring euthanasia due to morbidity were euthanized using $CO_2$ and then underwent complete necropsy. For each mouse, adrenal gland, bone, bone marrow, brain, heart, small and large intestine, kidney, liver, lungs, pancreas, spleen, testes, thymus, and any additional tumors or gross abnormalities identified were embedded and sectioned, and a single representative slide was stained with hematoxylin and eosin and examined by a trained veterinary pathologist (R.T.B.). The pathologist was blinded to the experimental condition of the animals (e.g. *Kdm6a* F1, control F1, *Kdm6a* F2, or control F2). When possible, the entire organ was included on the slide. The complete set of conditions identified in F1 and F2 mice was tabulated once all mice had undergone necropsy.

## Immunohistochemistry

All IHC was performed on the Leica Bond III automated staining platform. Anti-CD3 (A0452, clone F7.2.38, Dako, Santa Clara, CA) was run at 1:250 dilution using the Leica Biosystems Refine Detection Kit with EDTA antigen retrieval. Anti-CD20 (M0755, clone L26, Dako) was run at 1:500 dilution using the Leica Biosystems Refine Detection Kit with citrate antigen retrieval. Anti-VEGF (ab52917, clone EP1176Y, Abcam) was run at 1:100 dilution using the Leica Biosystems Refine Detection Kit with EDTA antigen retrieval. Anti-F4/80 (MCA497GA, clone CI:A3-1, Serotec, Hercules, CA) was run at 1:5000 dilution using the Leica Biosystems Refine Detection Kit with enzymatic antigen retrieval.

## Bone marrow sample collection

To collect bone marrow for flow cytometry analysis, RRBS, and RNA-seq, mice were euthanized by an overdose of carprofen (25 mg/kg) by intraperitoneal injection. The sternum was removed and fixed in 10% formalin for histological analysis. The spinal column, pelvic bone, and both femurs, fibulas, and tibias were stripped of muscle tissue and macerated in wash buffer (PBS + 2% FBS) using a mortar and pestle. All liquid was pipetted off of the remaining solid tissue and passed through a 100 micrometer (um) filter into a 50 mL Falcon tube, then spun down at 1200 rpm for 5 min at 4C. Supernatant was removed, and cells were resuspended in 10 mL red blood cell lysis buffer (#555899, Becton Dickinson, Mountain View, CA) and incubated for 5 min on ice. 20 mL wash buffer was added and the cells were passed through a 70 um filter into a fresh tube, then spun down again. The supernatant was removed, cells were resuspended in 20 mL wash buffer and passed through a 40 um filter into a fresh tube. Approximately 1 mL of this cell suspension was removed for DNA isolation for RRBS (see below). The remaining suspension was spun down one more time, then resuspended in freeze solution (90% FBS +10% DMSO), aliquoted to cryotubes and stored in liquid nitrogen.

## Flow cytometry

Peripheral blood, bone marrow, spleen and tumor cells were analyzed using the LSRII-Fortessa instrument (Becton Dickinson) using anti-mouse CD11b (clone M1/70, BioLegend, San Diego, CA),

anti-mouse Gr1 (clone RB6-8C5, BioLegend) and LIVE/DEAD Fixable Aqua Dead Cell Stain Kit (Life Technologies, Carlsbad, CA). Figures were prepared using FCSalyzer version 0.9.13.

## Western blotting

To collect germ cell-enriched mouse testis tissue, *Kdm6a* cKO and control male littermates were euthanized and testes transferred to 3 cm culture dish on ice, keeping individuals separate. The tunicae were removed and 450 ul cold collagenase solution (0.1% (w/v) hyaluronidase (#H3506, Sigma Aldrich, St. Louis, MO), 0.2% (w/v) collagenase (#C5138, Sigma Aldrich), 1:500 DNAse I (#07900, Stem Cell Technologies, Vancouver, Canada) in PBS was added. Testes tubules were teased apart using forceps for 7 min at room temperature. Liquid was removed and the sample was washed twice for 3 min in 450 ul wash solution (1:1000 DNAse I in PBS), with continued teasing. After the last wash, liquid was removed and tubules were resuspended in 700 ul trypsin solution (0.2% collagenase, 0.25% trypsin, 0.1 mM EDTA, 1:1000 DNAse I in water) and pipetted vigorously to break up clumps. Samples were shaken for 10–15 min at room temperature and then quenched with 700 ul Cosmic Calf Serum, and any remaining tissue chunks were allowed to settle. The cell suspension was transferred to a new tube and spun down at 3000xg, 4 min, 4C. The supernatant was removed and cells were resuspended in RIPA buffer containing protease inhibitor cocktail (sc-24948, Santa Cruz Biotechnology). The protein concentration was measured with Pierce BCA protein assay kit (#23225, Thermo Scientific). 30 ug of total protein was used for each blot and was incubated overnight with primary antibody against H3K27me3 (#07–449, Millipore Sigma) and GAPDH (#sc-32233, Santa Cruz Biotechnology). Blots were imaged on a FluorChem E System (ProteinSimple, San Jose, CA). Relative protein expression levels were quantitated using ImageJ and normalized to GAPDH. Blots were performed in triplicate for two biological replicates.

## Round spermatid collection

Dissociated testis cells were collected from *Kdm6a* cKO and control littermates as described above. The supernatant was removed and cells were resuspended in 1 mL cold resuspension solution (1% BSA in PBS). 2 ul DyeCycle Green (#V35004, Thermo Fisher, Waltham, MA) was added, and the cell suspension was mixed by inversion and then incubated for 30 min at 37C in the dark. Cells were then passed through a 40 ul filter. Round spermatids were recovered by flow cytometry using a FACSJazz (Beckton Dickinson) after sorting for cells with 1C DNA content and large size (to differentiate elongating from round spermatids). The purity of the cell population was verified by fluorescence microscopy (≥95% round spermatids) and by qPCR (*Figure 4—figure supplement 3*). qPCR primer sequences are listed in the Key Resources table.

## Sperm collection

Epididymal sperm for ChIP-seq and RRBS was collected by swim-up as follows: cauda epididymi were recovered from euthanized mice and cut 4–6 times on parafilm, then transferred to 6 cm culture dishes containing 5 mL of Donner's medium (135 mM NaCl, 5 mM KCl, 1 mM MgSO$_4$, 2 mM CaCl$_2$, 30 mM HEPES, 25 mM NaHCO$_3$, 20 mg/mL BSA, 1 mM sodium pyruvate, 0.53% (v/v) sodium DL-lactate), keeping tissue from each mouse separate. Epididymes were incubated at 37C for 1 hr with periodic gentle agitation, then passed through a 40 um filter, washed 1x in cold 0.45% NaCl to lyse any red blood cells and 1x in cold PBS. Sperm were resuspended in PBS, and 10 ul were removed for counting following standard procedures.

## Pyrosequencing

Pyrosequencing for three control sperm samples, three *Kdm6a* cKO sperm samples, three control F1 bone marrow samples, and three *Kdm6a* F1 bone marrow samples was performed at 13 loci by EpigenDx (Hopkinton, MA) according to the company's standard protocols. EpigenDx was blinded to tissue and experimental condition.

## Chromatin immunoprecipitation

### Round spermatids

Following isolation by flow cytometry, round spermatid samples were spun down and resuspended in 500 ul cold PBS, then fixed in 1% formaldehyde for 10 min at room temperature and quenched

with 2.5 M glycine for 5 min at room temperature. Samples were then washed twice in cold PBS, resuspended in 100 ul ChIP lysis buffer (1% SDS, 10 mM EDTA, 50 mM Tris-HCl [pH 8]), snap frozen in liquid nitrogen, and stored at −80C. For H3K4me1 ChIP, *Kdm6a* cKO and control samples (approximately $5 \times 10^5$ cells each, with samples from individual males kept separate) were thawed on ice and ChIP performed as previously described for round spermatid samples (*Lesch et al., 2016*). Briefly, samples were sonicated in a BioRuptor (Diagenode, Liege, Belgium) and then immunoprecipitated overnight at 4C, using 1 ug anti-H3K4me1 (ab8895, Abcam) or 1 ug anti-H3K27ac (ab4729, Abcam). The following morning, samples were incubated with Protein G Dynabeads (#10004D, Life Technologies) for 2 hr, washed, then eluted and reverse cross-linked. Following incubation with RNAse A and proteinase K, DNA was purified using a ChIP Clean and Concentrator kit (#D5201, Zymo Research, Irvine, CA). Spermatids from a single male were used to generate each ChIP library.

## Sperm

ChIP-seq in sperm was performed using a native ChIP protocol according to *Hisano et al. (2013)*. Briefly, sperm were resuspended in 1 mL cold PBS. 50 ul of 1M DTT was added and samples were incubated for 2 hr at room temperature. 120 ul 1M N-ethylmaleimide (#P4557, Sigma Aldrich) was added and the sample was incubated for another 20 min at room temperature. An aliquot was removed as a pre-MNase control, the sample was digested with 10 units of MNase (#10107921001, Sigma Aldrich) for 5 min at 37C, and 2 ul 0.5M EDTA was added to stop the digest. The chromatin solution was precleared for 1 hr with pre-blocked Protein G Dynabeads, then removed from the beads. 100 ul was set aside as a pre-ChIP control, and the remainder of the sample was incubated with 5 ug anti-H3K27me3 (ab6002, Abcam) overnight at 4C. The following day, chromatin samples were incubated for 8 hr with pre-blocked beads, then washed, eluted from the beads, and treated with RNAse A and proteinase K. DNA was purified using a Zymo ChIP Clean and Concentrator kit. Two biological replicates were prepared for each of the control and *Kdm6a* cKO genotypes. For each replicate, sperm from five males was pooled in a single ChIP experiment in order to recover enough histones for robust ChIP.

## Library preparation and sequencing

Both sperm and spermatid ChIP libraries were prepared using the Accel-NGS 2S Plus DNA Library Kit (#21024, Swift Biosciences, Ann Arbor, MI) according to the manufacturer's instructions, except that size selection was performed after (instead of before) PCR amplification. All libraries were sequenced on an Illumina HiSeq2500 with 40-base-pair single-end reads (Supplementary file 1).

## Data analysis

ChIP-seq reads were filtered using fastq_quality_filter from FASTX Toolkit version 0.0.14 with parameters –q 20 –p 80, and aligned to the mm10 mouse genome using bowtie version 1.2 with parameters –m 1 –k 1 –n 1 –l 40. We called sperm H3K27me3 peaks using MACS version 2.1.0 with parameters –-broad –-broad-cutoff 0.01 –-keep-dup 1 –-nomodel, and spermatid H3K4me1 peaks using MACS version 1.4.2 with parameters –p 1e-5 –-keep-dup 1 –-nomodel (*Zhang et al., 2008*). For sperm data, we used the pre-ChIP control sample as input. We also quantitated sperm H3K27me3 ChIP-seq signal in two kilobase tiles across the genome using methods from *Hisano et al. (2013)*. We scaled ChIP and input data to get reads per million, then subtracted input from ChIP signal for each tile and set any negative values to zero. To avoid damping the variation in signal between regions, we then re-scaled each dataset using the inverse of the reads-per-million scaling parameter originally used for the ChIP sample. Only those regions called as different in both replicates were used in downstream analyses. Log fold change (logFC) for H3K27me3 ChIP data was calculated by dividing the scaled, input-subtracted ChIP signal from *Kdm6a* cKO by the scaled, input-subtracted ChIP signal from control samples, and taking the log2 of the resulting quantity. Peak coordinates and tiled ChIP signal values are available at GEO under accession number GSE102313.

## Reduced representation bisulfite sequencing (RRBS)

### Sample collection and library preparation

Reduced representation bisulfite sequencing was performed on DNA prepared using the Qiagen DNEasy Blood and Tissue Kit (#69504, Qiagen, Hilden, Germany). Sperm were isolated by swim-up as described above and resuspended in 100 ul PBS. 100 ul buffer X2 (20 mM Tris-HCl [pH 8.0], 20 mM EDTA, 200 mM NaCl, 4% SDS, 80 mM DTT, 12.5 ul/ml Proteinase K from Qiagen DNEasy Blood and Tissue kit) was added and the sample was incubated for 2 hr at 56C on a shaking heat block. Samples were then treated with 2 mg/ml RNAse A for 2 min at room temperature. 200 ul buffer AL and 200 ul 100% ethanol were added, vortexing between each. Samples were transferred to DNEasy spin columns, and the remainder of the extraction was carried out according to the kit protocol. Aliquots of bone marrow, spleen, and liver tumor cells were spun down and resuspended in 200 ul PBS, and DNA was extracted according to the kit protocol. RRBS libraries were prepared using the NuGen Ovation RRBS Methyl-Seq System (#0353, NuGen, San Carlos, CA) according to the manufacturer's protocol, using 200 ng of DNA from each sample. Libraries were sequenced on an Illumina HiSeq2500 with 75 bp single-end reads and using 12 rounds of index sequencing. Equal numbers of *Kdm6a* cKO and control sperm samples, or *Kdm6a* F1 and control F1 bone marrow samples, were prepared and sequenced in parallel. Sperm and bone marrow libraries were prepared and sequenced on different days. Each RRBS library was derived from a single male (*Figure 4—source data 2*).

### Data analysis

We used trim_galore version 0.4.2 to remove adapter sequences from RRBS reads, trimmed diversity sequences using a script provided with the NuGen Ovation RRBS kit, and aligned reads to the mm10 genome using Bismark version 0.16.3 (*Krueger and Andrews, 2011*) and bowtie2 (*Langmead and Salzberg, 2012*). PCR duplicates were removed using the unique molecular identifiers (UMIs) added during library prep. Percent methylation at individual CpGs and in 100 bp tiles across the genome was called using the methylKit package in R, and only tiles covered by at least 10 reads were considered in further analyses (*Akalin et al., 2012*). Differential methylation between *Kdm6a* cKO and control sperm and between *Kdm6a* F1 and control F1 bone marrow was called using the calculateDiffMeth function in methylKit. To prevent confounding of bisulfite conversion by genomic variants between strains, we excluded all tiles containing known C > T or G > A variants between C57BL/6 and 129S4. When more than two tiles called as significant were less than 1 kb apart, one was selected at random to represent the genomic region, in order to avoid weighting a single region too heavily during characterization of associated features. *Figure 4D*, showing distribution of DMRs across the genome, was generated using the Phenogram tool (*Ritchie, 2012*; http://visualization.ritchielab.psu.edu/phenograms/document). Tables of percent methylation at individual CpGs are available at GEO under accession number GSE102313.

## RNA-seq

### Sample and library preparation

One vial of viably frozen bone marrow cells was thawed for each sample and washed once in PBS. One fifth of each sample ($2-6 \times 10^6$ cells per sample) was used for RNA extraction using the RNEasy Plus Mini Kit (#74134, Qiagen) according to the kit protocol. Libraries were prepared using the Illumina TruSeq RNA library prep kit (#RS-122–2001, Illumina, San Diego, CA) and sequenced on an Illumina HiSeq2500 with 40 bp paired-end reads. All RNA-seq libraries were prepared and sequenced in parallel.

### Data analysis

RNA-seq data was quantified using kallisto version 0.43.0 (*Bray et al., 2016*) with default parameters and with Ensembl build 85 transcripts (*Yates et al., 2016*) as the set of target sequences. Transcripts per million for individual transcripts from a single gene were summed to get one value per gene, and differentially expressed genes were called using DESeq2 in R (*Love et al., 2014*), after excluding immunoglobulin variable region transcripts and genes with total expression level <1 TPM across all samples. Genes were considered significantly differentially expressed for adjusted p-value<0.05.

Heatmaps were generated using the heatmap.2 function in the gplots package in R (*Warnes et al., 2016*). TPM tables are available at GEO under accession number GSE102313.

### Survival analysis

Kaplan-Meier curves were generated in R (*R Development Core Team, 2015*) using the package rms (*Harrell, 2016*). Hazard ratios and p-values for survival were calculated using a Cox proportional hazards model, using the R package survival (*Therneau, 2015*).

### Motif enrichment

Transcription factor binding motifs enriched in persistent hypermethylated DMRs were identified using AME (*McLeay and Bailey, 2010*) with the motif databases UniPROBE mouse (386 motifs) (*Badis et al., 2009*; *Berger et al., 2008*), JASPAR CORE vertebrates (519 motifs) (*Mathelier et al., 2016*) and human/mouse HT-SELEX (843 motifs) (*Jolma et al., 2013*). The full set of 100 bp tiles covered by at least 10 RRBS reads (263820 total) was used as the control set. Fisher's exact test was used to determine significance.

### Principal component analysis

Principal component analysis was carried out using the PCA function in the FactoMineR package in R (*Le et al., 2008*).

### Data availability

All sequencing datasets are available at GEO under accession number GSE102313.

## Acknowledgments

We thank P Nicholls, M Kojima, and M Mikedis for help with experiments, G Welstead and R Jaenisch for the *Kdm6a* mouse strain, M Goodheart for help with mouse procedures, and M Gehring, T Jacks, R Young, Y Stelzer, and K Stansifer for helpful discussions and advice. We thank the Whitehead Genome Technology Core for sequencing and the Koch Institute Histology Core for histological sample preparation and immunohistochemistry. We also thank Dana-Farber/Harvard Cancer Center in Boston, MA, for the use of the Specialized Histopathology Core, which provided histology and immunohistochemistry service. Dana-Farber/Harvard Cancer Center is supported in part by an NCI Cancer Center Support Grant # NIH 5 P30 CA06516.

## Additional information

### Funding

| Funder | Grant reference number | Author |
| --- | --- | --- |
| Howard Hughes Medical Institute | | David C Page |
| Burroughs Wellcome Fund | | Bluma J Lesch |
| Hope Funds for Cancer Research | | Bluma J Lesch |
| National Institutes of Health | 5K12CA087723-12 | Zuzana Tothova |
| Leukemia and Lymphoma Society | 3363-13 | Zuzana Tothova |
| American Society of Clinical Oncology | | Zuzana Tothova |
| American Society of Hematology | | Zuzana Tothova |
| National Institutes of Health | R01HL082945 | Benjamin L Ebert |
| National Institutes of Health | P01CA108631 | Benjamin L Ebert |
| Edward P. Evans Foundation | | Benjamin L Ebert |

| | |
|---|---|
| Gabrielle's Angel Foundation for Cancer Research | Benjamin L Ebert |
| Leukemia and Lymphoma Society | Benjamin L Ebert |

The funders had no role in study design, data collection and interpretation, or the decision to submit the work for publication.

### Author contributions

Bluma J Lesch, Conceptualization, Data curation, Formal analysis, Validation, Investigation, Visualization, Methodology, Writing—original draft, Writing—review and editing; Zuzana Tothova, Validation, Investigation, Writing—review and editing; Elizabeth A Morgan, Formal analysis, Validation, Writing—review and editing; Zhicong Liao, Formal analysis, Investigation, Visualization, Writing—review and editing; Roderick T Bronson, Investigation, Writing—review and editing; Benjamin L Ebert, Resources, Supervision, Methodology, Writing—review and editing; David C Page, Conceptualization, Resources, Supervision, Funding acquisition, Writing—original draft, Writing—review and editing

### Author ORCIDs

Bluma J Lesch (iD) https://orcid.org/0000-0002-6689-0240
David C Page (iD) http://orcid.org/0000-0001-9920-3411

### Ethics

Animal experimentation: All mice were maintained at the Whitehead Institute animal facility. Animal care, breeding, and experiments were conducted in compliance with the Animal Welfare Act and with the recommendations in the Guide for the Care and Use of Laboratory Animals of the National Institutes of Health. The protocol was approved by the Animal Care and Use Committee at the Massachusetts Institute of Technology (protocol #0714-074-17).

### Decision letter and Author response

Decision letter https://doi.org/10.7554/eLife.39380.056
Author response https://doi.org/10.7554/eLife.39380.057

# Additional files

### Supplementary files

• Transparent reporting form
DOI: https://doi.org/10.7554/eLife.39380.044

### Data availability

ChIP-seq, RRBS, and RNA-seq data generated in this study have been deposited in GEO under accession code GSE102313.

The following dataset was generated:

| Author(s) | Year | Dataset title | Dataset URL | Database and Identifier |
|---|---|---|---|---|
| Lesch BJ, Page DC | 2019 | Epigenetic profiling in Utx germline conditional knockouts and F1 offspring | https://www.ncbi.nlm.nih.gov/geo/query/acc.cgi?acc=GSE102313 | NCBI Gene Expression Omnibus (GEO), GSE102313 |

The following previously published datasets were used:

| Author(s) | Year | Dataset title | Dataset URL | Database and Identifier |
|---|---|---|---|---|
| ENCODE Consortium/Bing Ren | 2012 | H3K27ac ChIP-seq on 8-week mouse bone marrow | https://www.encodeproject.org/experiments/ENCSR000CCL/ | ENCODE, ENCSR000CCL |

| ENCODE Consortium/Bing Ren | 2012 | H3K27ac ChIP-seq on 8-week BMDM | https://www.encodeproject.org/experiments/ENCSR000CFD/ | ENCODE, ENCSR000CFD |
| ENCODE Consortium/Bing Ren | 2012 | H3K4me1 ChIP-seq on 8-week mouse bone marrow | https://www.encodeproject.org/experiments/ENCSR000CAG/ | ENCODE, ENCSR000CAG |
| ENCODE Consortium/Bing Ren | 2012 | H3K4me1 ChIP-seq on 8-week mouse BMDM | https://www.encodeproject.org/experiments/ENCSR000CFE/ | ENCODE, ENCSR000CFE |

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
