## [Decision Letter]

Thank you for sending your article entitled "Intergenerational epigenetic inheritance of cancer susceptibility in mammals" for peer review at *eLife*. Your article has been evaluated by two peer reviewers, and the evaluation has been overseen by a Reviewing Editor and Kevin Struhl as the Senior Editor.

Summary:

The manuscript by Lesch and colleagues describes how ablation of the H3K27me3 demethylase UTX in the mouse male germline leads to late onset phenotypes (tumour incidence) in genetically wild-type progeny. Because Utx is encoded on the X chromosome, male progeny from males carrying a germ-cell specific deletion of Utx are genetically wild-type and have normal zygotic expression of Utx. The authors show that such progeny have somewhat reduced long-term survival associated with increased incidence of a wide range of tumours. Tumour incidence appears to be potentiated in F2 progeny after passage through an Utx-deficient F1 male germline, but not through an intervening Utx-proficient germline. In terms of molecular correlates, the authors report altered H3K27me3 enrichment profiles in round spermatids and sperm, with sites of altered H3K27me3 also displaying increases in DNA methylation. To establish a possible link between the altered epigenetic profile in sperm with the later phenotypes in offspring, they conduct RNA-seq and RRBS in bone marrow of control and “cKO” offspring, both diseased and presymptomatic, identifying some overlap between differentially methylation in sperm and these tissues. Finally, they identify candidate transcription factors binding in “persistent DMRs”, and make a plausible case for their association with the phenotypes. While this is an interesting report, two expert reviewers have raised largely overlapping concerns that would need to be addressed in a detailed revision.

Essential revisions:

1) Breeding design, assessment of lifespan and tumor incidence:

Lifespan and tumor incidence is assessed in the F1 progeny from 3 Utx cKO males and 2 controls. Authors combine data from the three mutant "lineages" together, and likewise for controls. However, data (F1 generation and F2 mutant) should be presented for each individual lineage, as done in Figure 2—figure supplement 2 for F2 controls. This is essential to know what is the penetrance of the increased cancer susceptibility (i.e. is progeny of all males equally affected, or rates differ depending on the progenitor?). The relation between tumor incidence and DNA methylation alterations needs to be presented in the context of a pedigree relationship well. This last point is relevant since the authors suggest cKO fathers transmit altered epigenetic information to their offspring affecting their health.

F2 animals come from a single grandparent. Related to the point above, this is not sufficient to assess variability and significance of the observed phenotype. Furthermore, is this animal one of the three used to generate F1? Finally, is it valid to compare *Utx* F2 to control F1? Working on a mixed background requires rather more than less pedigrees to be followed.

The mouse model used in this study induces UTX deletion in spermatogenic precursor cells, many weeks before the formation of round spermatids and spermatozoa that were profiled in this study. Remarkable, the authors relate mechanistically the suggested changes in H3K27me3 in round spermatids and sperm to the alterations in DNA methylation observed in sperm. Disappointingly, the authors did not raise the question at what stage of spermatogenesis (proliferating spermatogonia, meiotic spermatocytes or haploid spermatids) UTX deficiency would alter H3K27me3 levels and as consequence would induce changes in DNA methylation. The genome wide distributions as observed in differentiated spermatids for H3K27me3 likely do not reflect the distributions in spermatogonia, a stage that is well known to be subject to de novo DNA methylation. Moreover, since demethylation independent functions have been reported (and cited) for Utx, the reported changes in DNA methylation may not mechanistically need to relate to H3K27me3.

Hence, it is conceptually not fully appropriate to derive mechanistic conclusions without studying germ cells from earlier stages, such spermatogonia. As such, the findings are correlations, and do not provide enough relevant insights into the mechanistic dependencies between the two chromatin marks. Therefore, the work would be significantly strengthened if the authors would include ChIPs in earlier germ cells, such as spermatogonia.

2) Molecular characterization:

With the analysis presented in Figure 3, it is very unclear whether H3K27me3 is actually altered and what the statistically significant changes in distribution of the mark are. A single replicate for ChIP-seq is insufficient given technical variability of the technique, hence it is essential to perform at least duplicates or triplicates. Likewise, individual replicates should also be presented and correlated (i.e. correlations between replicates of controls, of *Utx* cKOs, and control – mutant comparisons). Which regions are statistically significantly differently marked by H3K27me3 (H3K27me3 – input, without baseline correction) between mutant versus control when taking the two control and mutant samples as separate measures? These analyses need to presented. Finally, since the ChIP-seq experiment lack appropriate measures to obtain informative quantitative data (e.g. use of chromatin spike-ins to normalize for IP efficiencies, bar-coding of input chromatin), the authors need to be conservative in calling differentially marked regions.

A global increase in H3K27me3 in *Utx* cKO sperm and round spermids is further claimed based on Figure 3B. The quantification behind this figure is questionable given that input-normalized counts per tile seem not to be normalized by library size and could therefore be reflecting differences in total number of reads. To claim a global increase in H3K27me3 authors need to complement the data with a Western Blot. Similarly, authors refer to a potential mild spreading of the mark in the regions surrounding tiles with highest H3K27me3 signal (Figure 3D-E). This could reflect technical artifacts such as GC bias and not be biological, and it seems to not be observed in round spermatids (Figure 3—figure supplement 6).

By contrast, RRBS data has been performed with appropriate number of samples and tissue types. As with ChIP data, variability among controls and mutants in sperm and somatic tissues should also be presented. In addition, RRBS on somatic tissue of the F1 progeny should be correlated back to the corresponding father. Figure 4H shows that persistent DMRs are significantly closer to enhancers than other covered regions ("All") and, nonetheless, are a few thousand base pairs away on average. How do authors mechanistically relate relatively distant DMRs to altered enhancer function?

Finally, authors argue that a slight gain in DNA methylation at persistent DMRs might affect binding of CpG-sensitive TF such the identified ETS factors, and provide an example in Figure 5. However, the transcriptional effects are minor (10 dysregulated genes) and, despite showing increased DNA methylation levels of the CpG contining the ELK1 motif (Figure 5C), the expression of the gene is not significantly altered (Figure 5D). Greater transcriptional changes in the *Utx* F1 histiocytic sarcoma can be reflecting the signature of the tumor and be not related per se to the underlying causes. Thus, the example provided does not sustain the conclusions made by the authors.

3) Differential methylation in sperm and persistent DMRs:

Having an additional molecular correlate to “validate” the ChIP-seq data is important, and DNA methylation is an obvious one, given that DNA methylation and H3K27me3 enrichment are mutually exclusive in some genomic contexts, and methylation could provide a more “stable” modification into progeny. But it is puzzling that sites in *Utx* cKO sperm with loss and gain of H3K27me3 both present increased DNA methylation (Figure 3F). Do the authors have an explanation for this paradoxical effect?

The implication is that “persistent” DMRs between sperm and somatic tissues of the F1s are sites that have been resistant to epigenetic reprogramming in early development. The major barrier to persistence of gamete-derived methylation would be preimplantation epigenetic reprogramming, which sees substantial erasure of DNA methylation especially from sperm-derived chromosomes. Even given the authors' correct comment that DNA methylation gains are “partial” and therefore varying between sperm, it would seem a very feasible experiment, given the magnitude of the methylation effects, to conduct single blastocyst methylation analysis (RRBS?) to demonstrate whether there is indeed persistence and that it likely varies between embryos.

[Editors' note: further revisions were requested prior to acceptance, as described below.]

Thank you for resubmitting your work entitled "Intergenerational epigenetic inheritance of cancer susceptibility in mammals" for further consideration at *eLife*. Your revised article has been favorably evaluated by Kevin Struhl (Senior Editor), a Reviewing Editor, and two reviewers.

The manuscript has been improved but there are some remaining issues that need to be addressed before acceptance, as outlined below (in discussion, reviewer 1 has indicated that they support all comments provided by reviewer 2).

We still think that the H3K27me3 data should be reanalyzed as was suggested by the referees. The current analysis does not sufficiently take into account the (possibly technical) variability between samples. Comparing the results of these different contrasts to the DNA methylation data might result in a more balanced interpretation of the H3K27me3 data. For clarity I include below the detailed comments made by the reviewers.

*Reviewer #1:*

The authors improved the manuscript by incorporating several of the recommendations provided by the reviewers. Regrettably, the request for additional RRBS data in embryos to consolidate the original findings of epigenetic inheritance was not implemented.

The overall critical point to the study is the reproducibility of some the data. Based on the revision, several points remain open:

Figure 1—figure supplement 8: the tumor incidence is variable for offspring of control and mutant sires. It is highly recommended to discuss this more extensively in the manuscript. To increase clarity, the number of offspring animals per sire needs to be indicated in this supplementary figure.

Figure 2B: Is there any scientific reason to leave out the survival data of control F2 animals, which are to some extend the temporary controls of the *Kdm6a* F2 animals?

Figure 2C to 2F: it is unclear to this reviewer why the tumor incidence data of the *Kdm6a* F1 and F2 data is statistically related to and discussed to only the control F1 but not the control F2 data. The lack of contemporary "external" controls for the F2 data is pressing.

Figure 3B. The western blot data nicely demonstrates that H3K27me3 levels are increased in the *Kdm6a* germ line deficient testis samples. While this is a good confirmation of the mouse model, it does however not further strengthen the original argument of increased H3K27me3 occupancy levels in *Kdm6a* deficient sperm, for which western blot analyses in sperm should have been done.

Figure 3—figure supplements 2 and 3 demonstrate that the H3K27me3 data is noisy and that the levels of enrichment are variable between conditions and replicates. Currently, the authors have taken the approach to call regions differentially methylated when being differentially methylated in two pair-wise conditions. Given the high level of noise, a more stringent approach is necessary in which the variation between the two control samples and between the two mutant samples is taken into account as well. Moreover, when exchanging the control and mutant samples in the two pair-wise comparisons, does one observe the same regions to be enriched or depleted in H3K27me3 in the mutant over control?

The results of such differential analyses, testing several contrasts, will strengthen the interpretation of the results, particularly when compared to the DNA methylation data.

It is not really clear what the authors mean with the term "H3K27me3 flattening". Please explain better in the text.

For each of the DNA methylation figures, please indicate how many samples were used to generate the error bars.

*Reviewer #2:*

The authors have made modifications that address some of my concerns. I welcome the increased transparency of the results, for example, by displaying survival curves for F1s separated by sire.

There remains a problem in the ChIP-seq analysis (by not having included spike-in controls) and the interpretations offered both in the Results section and Discussion.

“Those tiles with the highest overall H3K27me3 signal exhibited a paradoxical loss of H3K27me3 in *Kdm6a* cKOs”. But this is exactly what you might expect to see if there is global increase in H3K27me3 owing to ablation of an H3K27me3 demethylase. A retention of H3K27me3 in regions from which it would normally be removed by *KDM6A* during spermatogenesis would lead to an “artificial” decrease in the H3K37me3 signal in ChIP-seq data at regions that normally retain it and are not sites of *KDM6A*-mediated demethylation.

“This effect implies flattening of H3K27me3 in individual sperm, or increased variability of H3K27me3 placement between sperm”. I do not follow the logic of this statement. In individual sperm there are single nucleosomes (if retained) at each position, so how does the signal get flattened in individual sperm? I think it is essential that the authors provide the alternative interpretation that the “flattening” could be the “technical” artefact of suppressing peak heights because of increased H3K27me3 signal genome-wide. This does not detract from the possibility that there is genuine “spreading” of the H3K27me3 signal in the cKOs, over regions from which it would normally be removed by the action of *KDM6A*.

“We conclude that loss of *KDM6A* impairs normal deposition of H3K27me3 during spermatogenesis.” Given the activity of *KDM6A*, this seems the wrong way to phrase it: “deposition” suggests an active process. Would be better phrased as “impairs the normal pattern of distribution of H3K27me3…”.

The Discussion opens with a statement about *Utx* ablation leading to “redistribution” of the H3K27me3 mark and “destabilisation of the H3K27me3 mark leaving regions ordinarily marked by H3K27me3 vulnerable to DNA methylation”. Of course, there are multiple mechanisms by which the observed effect on DNA methylation could have come about, but this does seem to be a rather odd and weak statement. The simplest model must be that ablation of UTX should leave intact regions of H3K27me3 that do not normally change during spermatogenesis, but would lead to increased H3K27me3 in regions in which H3K27me3 is normally turned over by the action UTX. This would tend to be seen as sites of ectopic H3K27me3 or perhaps broadening of the distribution.

There is an explicit statement in the Discussion, “Early in preimplantation development, when most DNA methylation is removed from the paternal genome, some of these hypermethylated regions resist reprogramming, such that methylation persists in somatic tissues.” As the authors have chosen not to do methylation analysis in preimplantation embryos, I recommend they should include the essential caveat that they have not directly shown that there is persistence of altered methylation through the reprogramming period, and other mechanisms of persistence are possible. This is necessary because a number of cases of intergenerational transmission do not depend on persistence of a methylation signal.

---

## [Author Response]

We thank the reviewers for their thoughtful comments. We have substantially revised and updated our manuscript based on their suggestions, and we believe it has significantly improved. We have added new data and analyses, and we have revised our discussion and interpretations. Specifically, we have made the following changes:

1) We have added survival analyses broken down by individual sire (Figure 1–figure supplement 3, Figure 2—figure supplement 2).

2) We have added summaries of tumor incidence broken down by individual sire (Figure 1—figure supplement 8).

3) We now represent the variability in our RRBS data in our summary plots (Figures 3G, 4J, Figure 3—figure supplement 6, Figure 4—figure supplement 4, Figure 4—figure supplement 5).

4) We now provide sample browser tracks of RRBS data of hypermethylated DMRs (Figure 3—figure supplement 6).

5) We have added analysis of the correlations between all individual ChIP-seq replicates (Figure 3—figure supplement 2).

6) We now include an analysis of significant differences in H3K27me3 for each control/mutant replicate separately, as well as for the combined datasets (Figures 3A, 3D, Figure 3—figure supplements 3-5).

7) In order to be more conservative in our conclusions about changes in H3K27me3, we now require that ChIP signal differs in each control/mutant replicate as well as in the combined ChIP data (Figure 3F, Figure 3—figure supplement 7).

8) We have clarified that the data presented in Figure 3B (now Figure 3A) is normalized for library size.

9) We have added Western blot data, with quantitation, demonstrating a global increase in H3K27me3 in testes of Utx cKO males (Figure 3B).

10) We now discuss possible mechanistic relationships between DMRs and nearby enhancers (subsection “Persistent *Kdm6a* DMRs overlap enhancers associated with tumorigenesis”).

11) We have updated our discussion of the effects of DNA methylation on ETS transcription factor binding in Utx F1s (Figure 5) in order to be more conservative in our conclusions (subsection “Persistent *Kdm6a* DMRs can alter transcription factor binding at enhancers”).

12) We now include new analyses and a more thorough discussion of our finding that hypermethylated DMRs are associated with regions of both increased and decreased H3K27me3 (subsection “Altered epigenetic profiles in *Kdm6a* cKO male germ cells”, Figure 3—figure supplement 7).

13) We have added additional references supporting high levels of DNA methylation in mammalian sperm, as suggested by the reviewers (subsection “Altered epigenetic profiles in *Kdm6a* cKO male germ cells”).

14) We have made the gene name formatting changes suggested by the reviewers (Introduction section).

15) The reviewers correctly note that round spermatids represent a late developmental stage relative to the timing of Utx deletion, and that genome-wide chromatin state in spermatids may not provide mechanistic insight into the immediate changes induced by Utx loss. We have therefore updated the manuscript to focus only on data from mature sperm, and revised our discussion to clarify that a developmental mechanism remains to be determined. We have removed the spermatid data from the manuscript.

We note that the hundreds of loci exhibiting DNA methylation changes in at least one animal in our dataset make pedigree analysis on a locus-by-locus basis statistically untenable. Instead, we have drawn probabilistic conclusions based on analysis of pooled data, in keeping with the statistical power analyses we performed before beginning our study. As noted above, our revised manuscript includes representations of the variability in DNA methylation changes among individual animals; keeping this variability in mind, we have focused our analysis on changes that are statistically robust across individual animals.

We agree with the reviewers that a mechanistic explanation for the correlations between Utx loss, H3K27me3 changes, and DNA methylation changes remains to be fully explored. As discussed in our previous correspondence, a full mechanistic understanding goes beyond the aim of the current manuscript. In general, the relationship between H3K27me3 and DNA methylation in mammalian cells is complex and poorly understood. While we were preparing our revisions, a study of polycomb-dependent epigenetic differences between naïve and primed embryonic stem cells (van Mierlo et al., 2019) reported that naïve mESCs exhibit flattening of H3K27me3 signals very similar to the effects we report in Utx cKO sperm, and that this redistribution also impacts DNA methylation. We now cite this study in our revised manuscript and include it in our Discussion section. We have also added analyses that support the relationship between H3K27me3 and DNA methylation in our proposed model (points 10 and 12 above), and we have revised our discussion to clarify why we favor this model over other mechanistic explanations of our data.

Our study was designed to demonstrate, for the first time, an intergenerational epigenetic effect on cancer phenotype in mammals. Our phenomenological data supporting this effect is a novel and significant achievement in a new and expanding field, and we also provide molecular correlates of the phenotype in sperm and in somatic tissue of offspring. The biology underlying this effect is complex and will require further study before it is completely understood; our report represents an important reference point from which to begin future exploration of this novel phenomenon.

[Editors' note: further revisions were requested prior to acceptance, as described below.]

The manuscript has been improved but there are some remaining issues that need to be addressed before acceptance, as outlined below (in discussion, reviewer 1 has indicated that they support all comments provided by reviewer 2).We still think that the H3K27me3 data should be reanalyzed as was suggested by the referees. The current analysis does not sufficiently take into account the (possibly technical) variability between samples. Comparing the results of these different contrasts to the DNA methylation data might result in a more balanced interpretation of the H3K27me3 data. For clarity I include below the detailed comments made by the reviewers.

We have reanalyzed our H3K27me3 data using the more stringent approach suggested, and find that it supports our original conclusions (see below).

Reviewer #1:

The authors improved the manuscript by incorporating several of the recommendations provided by the reviewers. Regrettably, the request for additional RRBS data in embryos to consolidate the original findings of epigenetic inheritance was not implemented.

As discussed in our previous response, RRBS data in embryos is beyond the scope of the current study.

The overall critical point to the study is the reproducibility of some the data. Based on the revision, several points remain open:Figure 1—figure supplement 8: the tumor incidence is variable for offspring of control and mutant sires. It is highly recommended to discuss this more extensively in the manuscript. To increase clarity, the number of offspring animals per sire needs to be indicated in this supplementary figure.

We discuss the variability in tumor incidence in the manuscript text in subsections “Reduced survival and increased tumor incidence in *Kdm6a* F1 compared to control F1 males” and “A model for epigenetic inheritance of cancer susceptibility”, and provide all tumor data in Figure 2—figure supplement 1. We have added the number of offspring animals per sire to Figure 1—figure supplement 8.

Figure 2B: Is there any scientific reason to leave out the survival data of control F2 animals, which are to some extend the temporary controls of the Kdm6a F2 animals?

Survival data for control F2 animals is provided in Figure 2—figure supplement 2.

Figure 2C to 2F: it is unclear to this reviewer why the tumor incidence data of the Kdm6a F1 and F2 data is statistically related to and discussed to only the control F1 but not the control F2 data. The lack of contemporary "external" controls for the F2 data is pressing.

In our experimental design, control F2 animals are the product of a *Kdm6a* cKO grandpaternal germline (Figure 2A), and are therefore not the appropriate statistical control for *Kdm6a* F2 animals. Control F1 animals are epigenetically wild type and are therefore offer a more appropriate comparison. Our data suggest that control F2 phenotypes are intermediate between *Kdm6a* F1 or F2 progeny and control F1s, suggesting a possible inherited epigenetic effect. However, our study was not powered to statistically support this secondary conclusion, so we do not address this matter in the text.

Figure 3B. The western blot data nicely demonstrates that H3K27me3 levels are increased in the Kdm6a germ line deficient testis samples. While this is a good confirmation of the mouse model, it does however not further strengthen the original argument of increased H3K27me3 occupancy levels in Kdm6a deficient sperm, for which western blot analyses in sperm should have been done.

We were unable to collect Western blot data for sperm because of the larger number of animals required. The whole testis Western blot data strongly supports the conclusion that *Kdm6a* knockout increases overall H3K27me3 levels during spermatogenesis, particularly in cells in the later stages of spermatogenic differentiation, which make up ~50% of all cells in the testis. In combination with our ChIP-seq results in sperm (Figure 3A), we believe our data support the conclusion that overall levels of H3K27me3 increase in sperm of the *Kdm6a* knockout.

Figure 3—figure supplements 2 and 3 demonstrate that the H3K27me3 data is noisy and that the levels of enrichment are variable between conditions and replicates. Currently, the authors have taken the approach to call regions differentially methylated when being differentially methylated in two pair-wise conditions. Given the high level of noise, a more stringent approach is necessary in which the variation between the two control samples and between the two mutant samples is taken into account as well.

We have redone this analysis after excluding any regions called as different in comparisons between the two control and two *Kdm6a* H3K27me3 datasets. The result supports our original conclusions; there is a small increase in effect size compared to the original analysis (Figure 3F).

Moreover, when exchanging the control and mutant samples in the two pair-wise comparisons, does one observe the same regions to be enriched or depleted in H3K27me3 in the mutant over control? The results of such differential analyses, testing several contrasts, will strengthen the interpretation of the results, particularly when compared to the DNA methylation data.

We have performed this analysis, and recovered the majority of regions identified in our original analysis (80% of the increased and 83% of the decreased regions; now presented in Figure 3—figure supplement 7).

It is not really clear what the authors mean with the term "H3K27me3 flattening". Please explain better in the text.

We have added an explanation of this term to the text (subsection “Altered epigenetic profiles in *Kdm6a* cKO male germ cells”).

For each of the DNA methylation figures, please indicate how many samples were used to generate the error bars.

Added.

Reviewer #2:

The authors have made modifications that address some of my concerns. I welcome the increased transparency of the results, for example, by displaying survival curves for F1s separated by sire.There remains a problem in the ChIP-seq analysis (by not having included spike-in controls) and the interpretations offered both in the Results section and Discussion.“those tiles with the highest overall H3K27me3 signal exhibited a paradoxical loss of H3K27me3 in Kdm6a cKOs”. But this is exactly what you might expect to see if there is global increase in H3K27me3 owing to ablation of an H3K27me3 demethylase. A retention of H3K27me3 in regions from which it would normally be removed by KDM6A during spermatogenesis would lead to an “artificial” decrease in the H3K37me3 signal in ChIP-seq data at regions that normally retain it and are not sites of KDM6A-mediated demethylation.

We agree with this assessment and have updated the text accordingly (subsection “Altered epigenetic profiles in *Kdm6a* cKO male germ cells”). We thank the reviewer for so clearly articulating this interpretation.

As a point of interest, we note that most previously published studies of *Kdm6a* knockouts have reported increased H3K27me3 signal at promoters, which differs from the loss of signal (whether real or relative) that we observe. Thus, although our results are “exactly what you might expect” following loss of a demethylase, they represent an unexpected effect in the context of this specific demethylase.

“This effect implies flattening of H3K27me3 in individual sperm, or increased variability of H3K27me3 placement between sperm”. I do not follow the logic of this statement. In individual sperm there are single nucleosomes (if retained) at each position, so how does the signal get flattened in individual sperm? I think it is essential that the authors provide the alternative interpretation that the “flattening” could be the “technical” artefact of suppressing peak heights because of increased H3K27me3 signal genome-wide. This does not detract from the possibility that there is genuine “spreading” of the H3K27me3 signal in the cKOs, over regions from which it would normally be removed by the action of KDM6A.

We agree and have updated this section of the text (see above).

“We conclude that loss of KDM6A impairs normal deposition of H3K27me3 during spermatogenesis.” Given the activity of KDM6A, this seems the wrong way to phrase it: “deposition” suggests an active process. Would be better phrased as “impairs the normal pattern of distribution of H3K27me3…”.

We have updated this phrasing.

The Discussion opens with a statement about Utx ablation leading to “redistribution” of the H3K27me3 mark and “destabilisation of the H3K27me3 mark leaving regions ordinarily marked by H3K27me3 vulnerable to DNA methylation”. Of course, there are multiple mechanisms by which the observed effect on DNA methylation could have come about, but this does seem to be a rather odd and weak statement. The simplest model must be that ablation of UTX should leave intact regions of H3K27me3 that do not normally change during spermatogenesis, but would lead to increased H3K27me3 in regions in which H3K27me3 is normally turned over by the action UTX. This would tend to be seen as sites of ectopic H3K27me3 or perhaps broadening of the distribution.

We have updated this section of the Discussion (subsection “A model for epigenetic inheritance of cancer susceptibility”).

There is an explicit statement in the Discussion, “Early in preimplantation development, when most DNA methylation is removed from the paternal genome, some of these hypermethylated regions resist reprogramming, such that methylation persists in somatic tissues.” As the authors have chosen not to do methylation analysis in preimplantation embryos, I recommend they should include the essential caveat that they have not directly shown that there is persistence of altered methylation through the reprogramming period, and other mechanisms of persistence are possible. This is necessary because a number of cases of intergenerational transmission do not depend on persistence of a methylation signal.

We have added this caveat (subsection “A model for epigenetic inheritance of cancer susceptibility”).